



# Algorithm Theoretical Basis for Ozone and Sulfur Dioxide Retrievals from DSCOVR EPIC

Xinzhou Huang[1] and Kai Yang[1]

[1]Department of Atmospheric and Oceanic Sciences, University Maryland, College Park, MD 20742, USA

**Correspondence:** Kai Yang (kaiyang@umd.edu)

**Abstract.**

Onboard the Deep Space Climate Observatory (DSCOVR), the first Earth-observing satellite at the L1 point (the first Lagrangian point in the Earth-Sun system), the Earth Polychromatic Imaging Camera (EPIC) observes the entire sunlit face of the Earth continuously. EPIC measures the solar backscattered and reflected radiances in ten discrete spectral channels, four of
5 which are in the ultraviolet (UV) range. These UV bands are selected primarily for total ozone ($O_3$) and aerosol retrievals based on heritage algorithms developed for the series of Total Ozone Mapping Spectrometer (TOMS). These UV measurements also provide sensitive detection of sulfur dioxide ($SO_2$) and volcanic ash, both of which may be episodically injected into the atmosphere during explosive volcanic eruptions. This paper presents the theoretical basis and mathematical procedures for the direct vertical column fitting (DVCF) algorithm used for retrieving total vertical columns of $O_3$ and $SO_2$ from DSCOVR EPIC.
This paper describes algorithm advances, including an improved $O_3$ profile representation that enables profile adjustments from multiple spectral measurements and the spatial optimal estimation (SOE) scheme that reduces $O_3$ artifacts resulted from EPIC's band-to-band misregistrations. Furthermore, this paper discusses detailed error analyses and presents inter-comparisons with correlative data to validate $O_3$ and $SO_2$ retrievals from EPIC.

## 15  1   Introduction

The Deep Space Climate Observatory (DSCOVR) was launched on the $11^{th}$ of February 2015, and after a 116-day journey maneuvered successfully into its Lissajous orbit around the first Earth-Sun system Lagrangian (L1) point, which is about $1.5{\times}10^6$ km from the Earth and located the between the Sun and the Earth on the ecliptic plane. At the L1 point, where the net gravitational forces equal the centrifugal force, DSCOVR orbits the Sun at the same rate as the Earth, staying close in line
along the Sun and the Earth and thus allowing the Earth-pointing EPIC to monitor the entire sunlit planet continuously.

The Earth Polychromatic Imaging Camera (EPIC) measures the solar backscattered and reflected radiances from the Earth using a 2-dimensional (2048×2048) charged-coupled device (CCD), recording a set of ten spectral images using different narrowband filters successively. While EPIC may observe the Earth continuously from the vicinity of the L1 point, only a





number of spectral image sets are taken in a day, limited by accessible contact windows of the two ground stations located in Wallops island (Virginia) and Fairbanks (Alaska). Currently, between 13 and 22 spectral image sets, recorded at a sampling rate of one set in every 110 minutes during boreal winter and every 65 minutes during boreal summer, are transmitted back to the ground stations in a day.

EPIC takes about six and a half minutes to complete an image set. The first in the set is the blue band (centered at 443 nm), which takes ∼2 minutes to complete the imaging at native resolution (2048×2048 pixels). The images of the nine remaining bands are recorded sequentially at a reduced resolution (1024×1024 pixels, achieved through an onboard average of 2×2 pixels), separating by a time cadence of ∼30 seconds between adjacent bands. Due to the Earth rotation and spacecraft jitter, each spectral image records a slightly different (i.e., rotated) sunlit hemisphere. As a result, the images of two different channels

appear to be displaced from each other, usually by a distance of about one to a few native pixels, depending on their observation time difference.

    Each native pixel has a ∼1 arc second or $2.778 \times 10^{-4}$ degree angular instantaneous field of view (IFOV), yielding a geometric ground footprint size of ∼8×8 $km^2$ at the image center of the sunlit disk. The effective footprint size is about 10×10 $km^2$, which is larger than the geometric one due to the effect of the optical point-spread function of the EPIC imaging system.

For a reduced resolution image (1024×1024 pixels), the effective central ground IFOV size is about 18×18 $km^2$, which is significantly smaller than the nadir footprints of some past and present satellite instruments that provided global ozone mapping from the low Earth orbit (LEO), such as the Total Ozone Mapping Spectrometer (TOMS, nadir pixel size 50×50 $km^2$) on a series of satellites, the Scanning Imaging Absorption Spectrometer for Atmospheric Cartography (SCIAMACHY, 60×30 $km^2$, Bovensmann et al. 1999) on ESA's ENVIronmental SATellite (ENVISAT), the Ozone Mapping and Profiler Suite Nadir

Mapper (OMPS-NM, 50×50 $km^2$, Flynn et al. 2014) on Suomi National Polar Partnership (SNPP), the Global Ozone Monitoring Experiment–2 (GOME-2, Callies et al. 2000; Munro et al. 2016) on Metop-A (40×40 $km^2$) and Metop-B (80×40 $km^2$). Though it is slightly larger than the nadir footprint of the Ozone Monitoring Instrument (OMI, 13×24 $km^2$, Levelt et al. 2006) on Aura and the OMPS-NM (17×13 $km^2$, Flynn et al. 2016) on NOAA-20, and much bigger than that of the TROPOspheric Monitoring Instrument (TROPOMI, 5.5×3.5 $km^2$, Veefkind et al. 2012) on the ESA Sentinel-5 Precursor (S5P), EPIC's spatial

resolution are sufficiently high to map small-scale $O_3$ natural variations and observe many volcanic emissions, from degassing to eruption.

    EPIC, combining moderate spatial resolution with high temporal cadences from the unique vantage point of L1, provides unprecedented Earth observations, from sunrise to sunset simultaneously (see Fig. 1). This synoptic (i.e., concurrent, globally unified, and spatially resolved) perspective is quite distinctive from satellite observations from an LEO or a geostationary Earth

orbit (GEO): LEO observations are often made within a narrow range of local time with a small number of samplings at a location per day, while GEO observations have limited spatial coverage, constrained to roughly 60° away from its position. The EPIC observations can have simultaneous co-located observations with measurements from any contemporaneous LEO and GEO platforms, allowing direct comparisons and synergistic use of data acquired from different perspectives. This overlapping feature has been exploited to calibrate some EPIC channels by matching its measured albedo values to those of OMPS-NM on

SNPP (Herman et al., 2017).





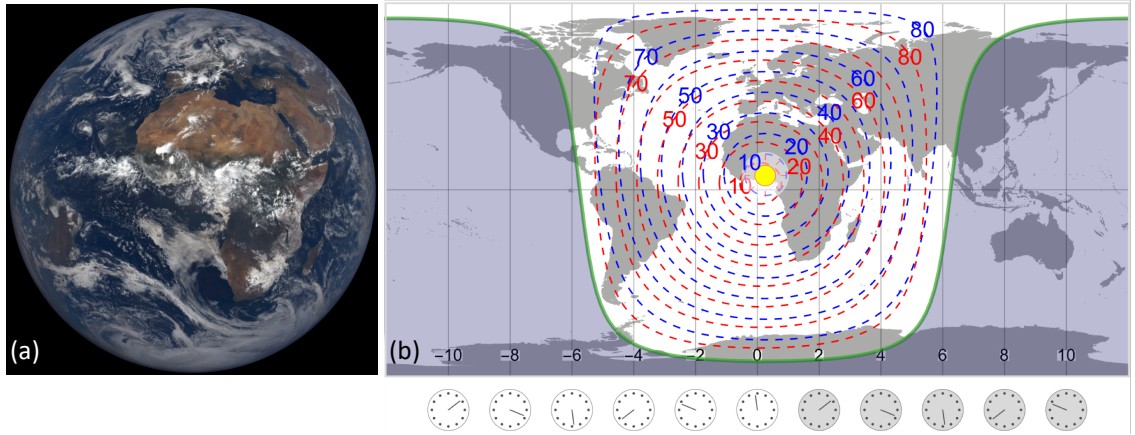

**Figure 1.** (a) Example of EPIC Field of View (FOV): EPIC earth image at 11:40:31 UTC on 4 September 2015. (b) Viewing and illumination angles are taken from FOV on the left. The subsolar point is marked on the map with a yellow dot. The area shaded with midnight blue is in the dark, i.e., without direct sunlight, while the unshaded area is the sunlit hemisphere, with sunrise on the left (west of subsolar point) and sunset on the right (east of subsolar point). Contours of solar zenith angles (SZAs, blue dashed lines) and viewing zenith angles (VZAs, red dashed lines), going from $10°$ to $80°$ with a step $10°$, are shown in the sunlit area. Note that the SZA ($\theta_s$) and VZA ($\theta_v$) of an EPIC IFOV have similar values and both angles increase as the IFOV moves from the center towards the edge of the sunlit disk.

The ten narrow bands of EPIC, spanning ultra-violet (UV), visible, and near-infrared wavelengths, are selected to yield diverse information about the Earth, from atmospheric compositions to surface reflectivity and vegetation. Four of the ten bands measure UV spectral radiances, which are used primarily for total ozone ($O_3$) retrievals. These UV bands also provide sensitive detection of sulfur dioxide ($SO_2$) and volcanic ash, both of which may be episodically injected into the atmosphere
5    during explosive volcanic eruptions.

This paper describes algorithm physics, model assumptions, mathematical procedures, and error analyses for the direct vertical fitting (DVCF) algorithm. We show examples to illustrate the high accuracy of $O_3$ and $SO_2$ retrievals achieved by applying the DVCF algorithm to spectral UV radiance measurements of DSCOVR EPIC. Lastly, we validate the DSCOVR EPIC $O_3$ and $SO_2$ through inter-comparisons with correlative data.

10    ## 2   Algorithm Physics

Algorithm physics is a term first used by Chance (2006) to denote the physical processes contributing to the spaceborne measurement of radiance spectra. A measured radiance $L_m$ (in units of $W \cdot sr^{-1} \cdot m^{-2} \cdot nm^{-1}$) from space consists of sunlight photons within a narrow spectral range (typically < 2 nm), specified by the instrument spectral response function $S$ (ISRF,

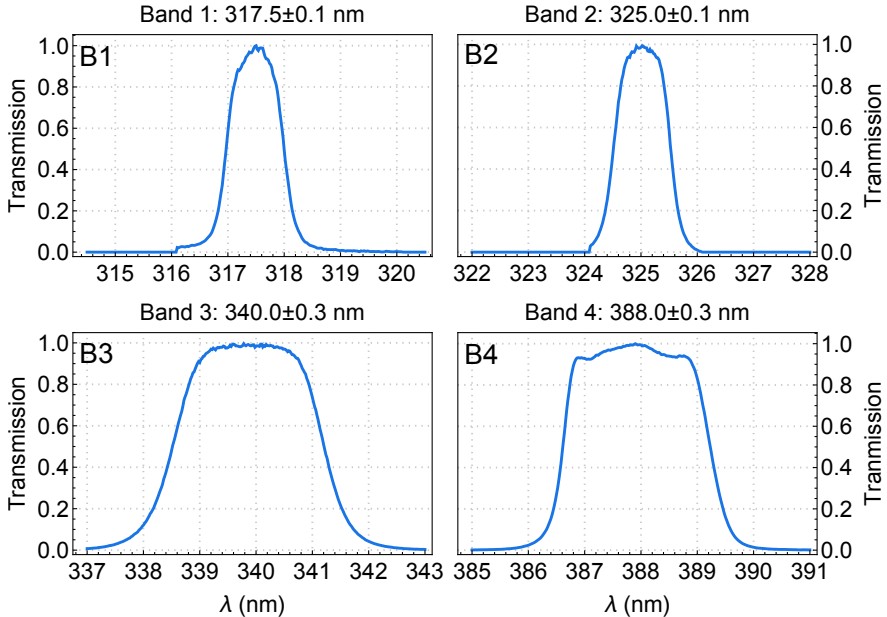

**Figure 2.** Filter transmission functions for the four EPIC UV channels. The widths are ∼1 nm for EPIC bands 1 and 2, similar to those for TOMS and OMPS-NM. Note that the filter transmissions as functions of wavelength are measured in the air (see Figure 1 in Herman et al. 2017). Here we have converted the wavelength in the air to wavelength in vacuum using the formula of Edlén (1966). The filter values are normalized to 1 at band centers (noted on top of each panel with uncertainty).

e.g., EPIC UV filter transmissions shown in Fig. 2), and is modeled as

$$L_M = \frac{\int S(\lambda) I_{TOA}(\lambda) F(\lambda) d\lambda}{\int S(\lambda) d\lambda}, \tag{1}$$

where $F(\lambda)$ (in units of $W \cdot m^{-2} \cdot nm^{-1}$) is the monochromatic spectral solar irradiance, and $I_{TOA}(\lambda)$ the sun-normalized monochromatic top-of-the-atmosphere (TOA) radiance (in units of $sr^{-1}$) for a wavelength $\lambda$ (in units of nm). The sun-normalized measured radiance $I_M$ for a spectral band is defined as $I_M = L_M/F_M$, where $F_M = \int S(\lambda) F(\lambda) d\lambda / \int S(\lambda) d\lambda$, and the $\lambda$ integrations in these equations are performed over the valid range of the ISRF $S$ for the spectral band. Hereafter we drop 'sun-normalized' when referring to $I_M$, which is simply called measured radiance. Quantities for a spectral band are flux-weighted bandpass averages to account for the differential contributions from individual wavelengths within the bandpass. Without loss of generality, $I_{TOA}(\lambda)$ and other spectral-dependent quantities are hereafter used to denote flux-weighted bandpass averages, with $\lambda$ representing the characterized wavelength of the spectral band.

To reach a sensor at TOA, sunlight photons are either back-scattered by air molecules or particles or reflected by the underlying Earth surface. As these photons traverse through the atmosphere along many possible optical paths connecting the Sun to the sensor, they may be absorbed by the underlying surface or by some atmospheric constituents, such as trace gases (e.g. $O_3$



and $SO_2$) and light-absorbing particles (e.g. dust and smoke). The photons that complete the journey carry information about atmospheric absorbers along their paths. The accumulation of photons from each contributing path yields the TOA radiance, which may be modeled with radiative transfer (RT) simulation if the properties of surface reflection and atmospheric absorption and scattering are known explicitly. The ability to model the TOA radiance accurately is the prerequisite for interpreting the observations and relating the gas absorptions with TOA radiance measurements.

We describe next the characteristics of UV photon sampling of the atmosphere, and the construction of surface and atmospheric models to enable proper simulation of the photon sampling of the atmosphere. Dividing the atmosphere into infinitesimal thin layers, the quantity that specifies the photon sampling is the mean path length of photons traversing through a layer. This mean path length normalized by the geometric thickness of the layer is the local or altitude-resolved air mass factor (AMF, $m_z$). The proper simulation of photon sampling requires that the modeled mean path length through each layer closely matches that in the actual observing condition.

In theory, a TOA radiance, $I_{TOA}$, depends on the viewing-illumination geometry, the optical properties of the atmospheric constituents (both absorbers and non-absorbers), and their amounts and vertical distributions, as well as on the reflective properties of the underlying surface. For a wavelength $\lambda$, $I_{TOA}$ can be expressed as the sum of two contributions,

$$I_{TOA} = I_a + I_s, \tag{2}$$

where $I_a$ consists of solar photons scattered once or more by molecules and particles in the atmosphere without interacting with the underlying surface, and $I_s$ are solar photons reflected at least once or multiple times by the underlying surface.

## 2.1 Path Radiance

$I_a$ is also known as the atmospheric path radiance, i.e., photons backscattered to the sensor along a path without any intersection with the underlying surface. Conceptually it is the accumulation of TOA photons that are last backscattered toward the sensor along the line of sight from atmospheric layers at different levels of extinction optical depths. Algebraically it is expressed as the path integration of virtual emission $J(t)$ (Dave 1964) in the direction specified by the view zenith angle ($\theta_v = \cos^{-1}\mu$), attenuated ($e^{-t/\mu}$) by atmospheric scattering and absorption, over the extinction optical depth $t$ along the path of line of sight from the top ($t = 0$) to the bottom ($t = \tau$) of the atmosphere:

$$I_a = \int_0^\tau J(t)\ e^{-t/\mu}\ \omega(t)\ dt/\mu. \tag{3}$$

The source of virtual emission, $J(t)$, consists of all the photons scattered towards to the sensor, including photons of the direct solar radiation being scattered once only and photons of diffuse radiation (i.e. photons scattered to level $t$) being scattered once more at $t$. The strength of the virtual emission of a thin layer at $t$ is proportional to its scattering optical thickness, which is equal to the product of the layer total optical thickness ($dt$) and the single scattering albedo $\omega(t)$ (defined as the ratio of layer scattering optical thickness over the layer total optical thickness). Here we use $\Psi(t) = J(t)e^{-t/\mu}\omega(t)/\mu$ to represent the radiance contribution per unit optical thickness to $I_a$ from a layer at $t$. Eq. (3) describes how the solar photons sample the

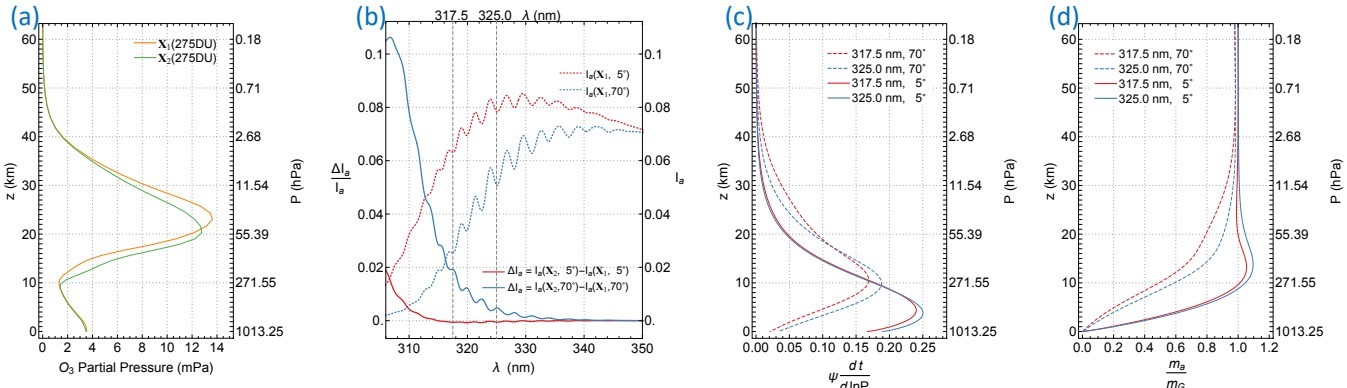

**Figure 3.** Sample results from RT simulations for a molecular atmosphere with $O_3$ profiles $\mathbf{X}_1$ and $\mathbf{X}_2$ in panel (a). Both $\mathbf{X}_1$ and $\mathbf{X}_2$ are mid-latitude zone ($30° \leq$ latitude $\leq 60°$) climatological $O_3$ profiles with the same total vertical column of 275 Dobson units, where 1 DU = $2.69 \times 10^{16}$ molecules/cm$^2$. RT simulations are performed for two viewing-illumination geometries: 1) low zenith angles, $\theta_s = \theta_v = 5°$ and relative azimuthal angle (RAA), $\phi = 45°$ and 2) high zenith angles, $\theta_s = \theta_v = 70°$ and $\phi = 45°$. (b) Path radiances $I_a(\mathbf{X}_1)$ for the low and high zenith geometries, and their fractional changes ($\Delta I_a / I_a$) when $O_3$ profile is changed to $\mathbf{X}_2$. (c) Normalized RCFs, $\psi$ for EPIC bands 1 and 2. Here $\psi(t)$ is converted into $\psi(\ln P)$ by the multiplication of factor $dt/d\ln P$. (d) Mean photon path lengths ($m_a$) of EPIC bands 1 and 2 as functions of altitude $z$ for the low and high zenith geometries, normalized by the respective geometric air mass factors, $m_G$.

atmosphere from top to bottom and how atmospheric absorption is directly imprinted (via the attenuation $e^{-t/\mu}$) on the path radiance.

A path radiance $I_a$ for a molecular (i.e., an aerosol- and cloud-free) atmosphere with absorption from trace gases can be accurately determined with RT simulations. For example, the path radiances for the low and high zenith angle geometries (see Fig. 3b) are calculated with a vector RT code (e.g., TOMRAD, Dave 1964, or VLIDORT, Spurr 2006) as a function of wavelength for a molecular atmosphere with the $O_3$ profile $\mathbf{X}_1$ in Fig. 3a, and the corresponding radiance contributions to the path radiances at EPIC bands 1 and 2 are shown in Fig. 3c. The radiance contribution function (RCF) for a wavelength in the UV range (300 – 400 nm) is determined by Rayleigh scattering and absorption by trace gases (primarily $O_3$). $O_3$ is ubiquitous in the atmosphere, with the bulk of it located in the stratosphere (e.g., Fig. 3a or Fig. 11), and its absorption cross-sections $\sigma(O_3)$ increase rapidly with shorter wavelengths in the UV range (see Fig. 13). Rayleigh scattering, whose cross-sections are proportional to $\frac{1}{\lambda^4}$, also increase with shorter wavelength. The strong $O_3$ absorption and large Rayleigh cross-sections at short wavelengths greatly reduce the number of solar photons reaching the lower atmosphere. Conversely, at longer wavelengths, weaker $O_3$ absorption and smaller Rayleigh cross-sections allow more solar photons to reach the lower atmosphere where higher air density increases the intensity of backscattering. Similar to the effect of reducing wavelength, lengthening the slant path (by increasing solar or viewing or both zenith angles) would enhance ozone absorption and Rayleigh scattering along the slant path, raising the altitude profile of RCF. These spectral and angular characteristics of RCF are illustrated in Fig. 3c, which shows the normalized RCFs ($\psi = \Psi / I_a$) of EPIC bands 1 and 2 for two different observation geometries and a mid-latitude $O_3$ profile labeled as $\mathbf{X}_1$ in Fig. 3a. The results in Fig. 3c show that at longer wavelengths and lower zenith angles, path radiance





contains more photons that are backscattered from the lower atmosphere. The RCF peak reaches ~4 km altitude for band 2 at 5° zenith angle, while at shorter wavelength and higher zenith angle, the RCF peak moves to the higher altitude, and it rises to ~10 km for band 1 at 70° zenith angle. The shifting shapes of RCF shown in Fig. 3c illustrate the changes in the photon sampling of the atmosphere with different wavelengths and zenith angles. The rising RCF peak position signifies diminishing sensitivity to absorptions below the peak while favoring those above it.

The measurement sensitivity to a thin molecular absorber layer is equal to the product of the absorption cross-sections ($\sigma$) and the mean path length ($m_a$) of photons passing through the layer, where $m_a = -\partial \ln I_a / \partial \tau_z$ and $\tau_z$ is the absorption optical depth at the layer center altitude $z$. Note that the photon path length is equal to the geometric AMF, $m_G = 1/\cos(\theta_s) + 1/\cos(\theta_v)$, for a plane-parallel atmosphere if there is no scattering. Figure 3d shows the mean optical path lengths of EPIC bands 1 and 2 as a function of altitude for the low and high zenith viewing-illumination geometries, showing that $m_a$ decreases rapidly as the layer descends nearing the surface due to fewer photons reaching the lower atmosphere while $m_a$ approaches $m_G$ as the layer rises towards TOA due to fewer path altering scatterings resulted from lower air density. In the upper troposphere and lower stratosphere (UTLS), $m_a$ of the low zenith geometry usually exceeds $m_G$ due to a significant fraction of photons undergo multiple scattering below and within UTLS, while $m_a$ of the high zenith geometry drops continuously from TOA down to the surface in the case when the RCF peak is sufficiently high that fewer multiple scatterings contribute to the path radiance. In general, the mean path length $m_a$ is shorter for a wavelength with stronger $O_3$ absorption, which reduces the number of photons reaching the lower atmosphere. The variation of $m_a$ with a changing altitude signifies the path radiance dependence on the absorber profile. The path radiance fractional change due to profile change ($\Delta \mathbf{X} = \mathbf{X}_2 - \mathbf{X}_1$) can be expressed as

$$\frac{\Delta I_a}{I_a} = \frac{I_a(\mathbf{X}_2) - I_a(\mathbf{X}_1)}{I_a(\mathbf{X}_1)} = -\int_0^\infty \Delta X(z)\, \sigma(T_z)\, m_a\, dz, \tag{4}$$

where $T_z$ is the atmospheric temperature and $X_1(z)$ and $X_2(z)$ are absorber concentration at altitude $z$. Figure 3b illustrates the change in path radiance caused by a $O_3$ profile change while keeping its total vertical column the same: lowering the $O_3$ profile (e.g., $\mathbf{X}_1$ to $\mathbf{X}_2$ in Fig. 3a) tend to increase the path radiance. Path radiance changes more with shorter wavelengths at higher zenith angles, thus becoming more sensitive to the shape of the $O_3$ profile. At low zenith angles, the change may have the opposite sign of the change at large zenith angle for certain wavelengths (e.g., the changes plotted as red solid lines for $\lambda > 316$ nm in Fig. 3b), but the magnitude of change is much smaller, indicating the path radiances under these conditions are primarily functions of total columns, since they are less sensitive to the profile shapes. The differential responses of the spectral path radiance to profile changes imply that more than one piece of information about $O_3$ may be contained in the multi-spectral measurements. Retrieval constrained by multi-spectral radiances instead of a single spectral band may achieve a more accurate $O_3$ measurement.

## 2.2  Surface Reflection

The path radiance $I_a$ includes backscattered photons that are independent of the underlying surface, while the surface contribution to TOA radiance, $I_s$ (referred to as surface radiance hereafter), consists of photons reflected once or more from the surface.


For a molecular atmosphere bounded by a surface with well-characterized optical reflection properties, the surface radiance $I_s$ can be accurately predicted with RT modeling. For a Lambertian surface, which reflects radiation isotropically independent of the incident direction, the surface radiance $I_s$ can be expressed as (Dave, 1964)

$$I_s = \frac{T_{\downarrow} r_s T_{\uparrow}}{1 - r_s S_b},\tag{5}$$

where $r_s$ is the reflectance or albedo of the Lambertian surface, $T_{\downarrow}$ is the total (direct and diffuse) transmittance from the Sun to the surface along the direction of incoming solar irradiation and $T_{\uparrow}$ from the surface to the TOA along the viewing direction, and $S_b$ is the atmospheric spherical albedo, which is the fraction of the reflected radiation backscattered from the overlaying atmosphere to the surface. The surface contribution from the Lambertian surface, $I_s$, may be described as the once-reflected radiance ($T_{\downarrow} r_s T_{\uparrow}$), enhanced by the series of interactions: backscattering from the overlaying atmosphere and reflection from

the underlying surface, which are accumulated to produce the amplification factor $1/(1 - r_s S_b)$.

The reflection property of a surface is represented by a bidirectional reflectance distribution function (BRDF), which specifies the angular distribution of reflected radiance as a fraction of directional incident spectral irradiance. Field measurements (Brennan and Bandeen, 1970) demonstrate that the reflection from natural surfaces (such as cloud, water, and land surfaces) are anisotropic in the UV, exhibiting different apparent reflectances when viewed from different directions. For instance, a

water surface looks bright when viewed from the direction near the specular reflection, but is much darker outside the glitter (e.g., see Fig. 4a). Here the apparent reflectance is the Lambertian-Equivalent reflectivity (LER), i.e., the isotropic reflectance $r_s$ that reproduces the radiance $I_s$ from a surface with an anisotropic BRDF at a viewing-illumination geometry. This LER is also referred to as geometry-dependent surface LER (GLER) to indicate its dependence on the viewing-illumination geometry.

Reflection of UV sunlight from natural surfaces has long been measured by instruments onboard satellites in sun-synchronous

polar orbits (e.g. Eck et al., 1987). Since BRDFs for most natural surfaces (except for water surfaces) have not been adequately characterized in the UV, satellite measurements provide scene reflectivities that are quantified with LERs at wavelengths in the range of weak gaseous absorption. To derive LER $r_s$ from a measured radiance $I_M$, the atmospheric path radiance $I_a$, transmissions $T_{\downarrow}$ and $T_{\uparrow}$, and reflectance $S_b$ for a spectral band are calculated for a molecular atmosphere and the inversion of Eq. (5) yields

$$r_s = \frac{I_s}{T_{\downarrow} T_{\uparrow} + S_b I_s},\tag{6}$$

where $I_s = I_M - I_a$. A vast majority of scene LERs derived from satellite observations contain contributions from scattering from clouds or aerosols or both (see section 2.3 for their treatment). To characterize reflective properties of natural surfaces, many investigations have devoted to creating global LER climatologies by selecting gridded LERs that are minimally affected by clouds or aerosols from the repeated observations over a period of time (typically a calendar month). These climatologies

include spectral surface LER databases constructed from the TOMS radiance measurements between 340–380 nm from 1978–1993 (Herman and Celarier, 1997), GOME-1 between 335–772 nm from 1995–2000 (Koelemeijer, 2003), SCIAMACHY between 335–1670 nm from 2002–2012 (Tilstra et al., 2017), OMI between 328–499 nm from 2005–2009 (Kleipool et al., 2008), and GOME-2 between 335–772 nm from 2007–2013 (Tilstra et al., 2017). Inter-comparisons of these spectral LERs



from different satellite missions show good agreement among corresponding measurements (Tilstra et al., 2017) despite differences in observation time periods, viewing-illumination geometry, and footprint size. For a location on Earth, its surface is usually observed at nearly the same local solar time from a sun-synchronous orbit, thus the sampling of its surface BRDF is limited to a small range of SZAs. Furthermore, the selection of cloud- and aerosol-free LERs tends to favor low LER values,

thus likely excluding the LERs at high VZAs. LER values of natural surfaces tend to be quite close when SZAs fall within a small range and large VZAs are excluded, hence these LER climatologies are presented as independent of viewing-illumination geometry. The low LER sensitivity to varying viewing-illumination geometry (within limited ranges of SZA and VZA) indicates that natural surfaces (excluding glittering water surface) have weak anisotropy and can be treated as Lambertian surfaces. These climatological data reveal that the surface LER in the UV for snow- and ice-free areas vary within the range of 0.02–0.1

for most land and (off-glint) water surfaces, except for a few places on Earth, such as the Saharan desert and the salt flat in Bolivia, where surface LERs may exceed 0.1. These low surface LER values derived from satellite observations have been validated in field experiments (Coulson and Reynolds 1971; Doda and Green 1980, 1981; Feister and Grewe 1995), which have found that the spectral reflectances of natural surfaces, such as the open ocean, forest, grassland, and desert, fall within the same range of satellite LER measurements. These field experiments have also demonstrated that the spectral reflectances

of natural surfaces vary slowly and smoothly with changing wavelengths. The spectrally smooth GLER of natural surfaces permits accurate estimation of GLER within the UV range with measurements at two or more wavelengths, and specifically, the extrapolation of GLERs determined at the long (weak $O_3$ absorption) wavelengths to estimate the GLERs at short (strong $O_3$ absorption) wavelengths.

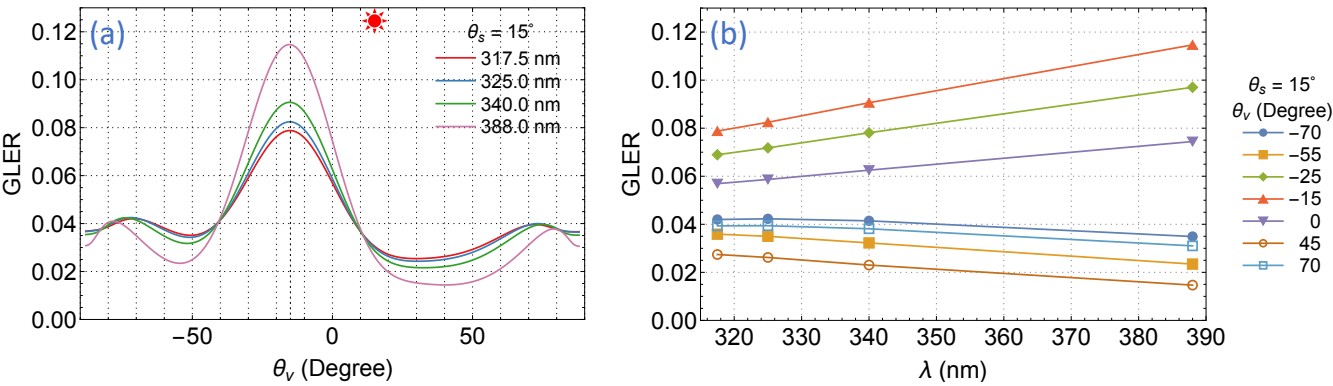

**Figure 4.** Apparent reflectances of an ocean surface, described by a Cox-Munk BRDF (Cox and Munk 1954a, b) for a wind speed of 6 m/s, viewed along the plane of incidence with the Sun at a zenith angle of $\theta_s = 15°$. (a) GLER at four EPIC UV bands vs viewing zenith angle $\theta_v$. Here positive $\theta_v$ denotes $\phi = 0°$ and negative $\theta_v$ for $\phi = 180°$. (b) GLER at several viewing zenith angles vs. wavelength $\lambda$.

Based on the reflective characteristics of natural surfaces described above, the forward model for retrieval treats the reflec-

tions from a surface as Lambertian, whose reflectance is determined from the radiance measurement of the spectral band with weak gaseous absorption or is extrapolated from the weak to the strong absorption band. We use the reflection from an ocean

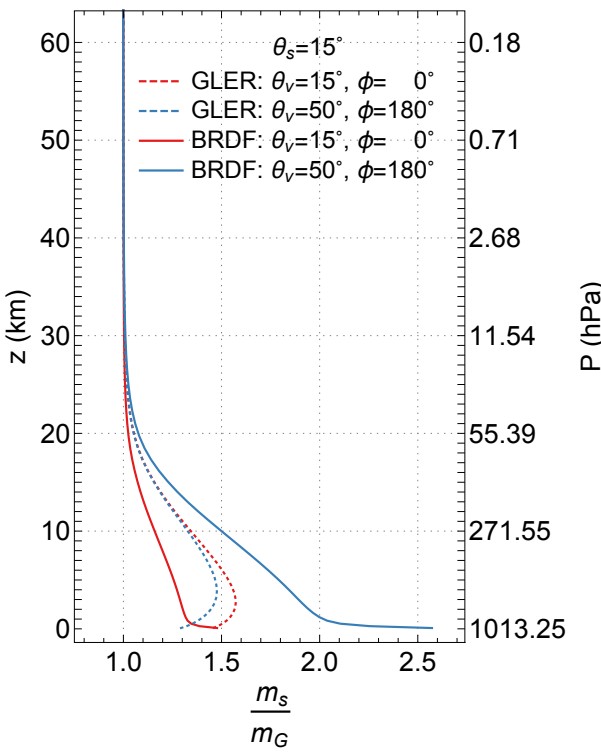

**Figure 5.** Mean path lengths ($m_s$) of EPIC band 1 reflective photons from an ocean surface (with the same BRDF described in Fig. 4) and its Lambertian equivalent surfaces. Here the mean path lengths $m_s$, normalized by the respective geometric air mass factors ($m_G$), are plotted as functions of altitude $z$ for two viewing-illumination geometries: one view from the direction of specular reflection, $\theta_v = 15°, \phi = 180°$, and the other at $\theta_v = 50°, \phi = 0°$, while the Sun at $\theta_s = 15°$ for both geometries.

surface as an example to illustrate the success and deficiency of the isotropic surface treatment and the GLER extrapolation, since a water surface is likely the most anisotropic surface encountered in satellite remote sensing. Figure 4a displays the GLERs of an ocean surface at the four EPIC UV bands as a function of VZA along the incident plane with the Sun at $\theta_s = 15°$. Viewing in the specular direction ($\theta_v = 15°$ and $\phi = 180°$), the GLER decreases with longer wavelengths but the reverse is

5   true when viewing in directions $\sim25°$ or greater away on either side of it. In other words, the reflection appears to be less anisotropic at shorter wavelengths. This is due to less direct beam, thus more diffuse radiation (resulted from more photons are Rayleigh scattered by air molecules) at the shorter wavelengths. While the reflection of a direct beam yields anisotropic outgoing radiation according to the BRDF, the diffuse radiation impinges on the surface from every possible direction of the hemisphere above, usually resulting in a much less anisotropic reflected radiation, which follows the angular distribution spec-

10   ified by the hemispherically averaged BRDF. Figure 4b shows the spectral dependence of GLER on wavelength, illustrating that linear extrapolation of GLER at longer wavelengths (340.0 nm and 388.0 nm) yields highly accurate GLER estimations at shorter wavelengths (317.5 nm and 325.0 nm), usually with errors much less than 1%.



The Lambertian surface treatment enables an accurate estimation of the surface radiance $I_s$ without the knowledge of the actual BRDF, provided that the GLERs estimated at some (usually the weak absorbing) wavelengths can be extended (linearly extrapolated) to other wavelengths accurately. However, the paths traversed by photons reflected from a Lambertian surface differ from those from an anisotropic one, as illustrated in Fig. 5, which displays the mean optical path lengths, $m_s = -\partial \ln I_s / \partial \tau_z$,

of EPIC band 1 as a function of altitude for two viewing-illumination geometries. As shown in Fig. 5, the path lengths differ the most just above the surface, but the difference decreases with higher altitudes due to less course-altering atmospheric scattering resulting from lower air density and vanishes around 25 km above the surface. Thus the Lambertian treatment of an anisotropically reflective surface may introduce an error, called the AMF error, in accounting for atmospheric absorption due to the difference in the photon sampling of the atmosphere. Since this difference is larger in the lower troposphere, but becomes

negligible in the stratosphere, implying that the effect of anisotropic reflection, i.e., the BRDF effect, has a larger impact on the quantification of trace gas absorption in the troposphere, but a smaller one for trace gases in the stratosphere. Because the bulk $O_3$ ($\sim 90\%$) is located in the stratosphere, the Lambertian treatment does not introduce a significant AMF error in total $O_3$ absorption.

As described above, UV reflectivities for most natural surfaces are quite low (GLER < 0.1), therefore the surface contri-

butions $I_s$ are typically much smaller than (< 10% at 317.5 nm) the path radiance $I_a$ (see Fig. 6). In modeling a measured radiance $I_M$, an error in surface radiance $I_s$ is compensated for with the path radiance $I_a$. The uncertainty of extrapolated GLER is usually less than 1%, corresponding to a less than 1% error in $I_s$, hence less than 0.1% error in the path radiance $I_a$. Furthermore, the AMF error due to the Lambertian treatment of an anisotropic surface is insignificant, since the combined mean photon path lengths,

$$m_z = -\partial \ln I_{TOA} / \partial \tau_z = (I_a m_a + I_s m_s)/I_{TOA}, \tag{7}$$

contain minor contributions from surface radiance $I_s$.

Natural surfaces with high UV reflectivities (GLER >0.2) are surfaces covered with snow or ice or both. The highest GLER values are found over Antarctica and Greenland, where typical GLER values are higher than 0.9, as shown in Fig. 7). Figure 7 shows sample results of a climatological GLER database for Antarctic ice constructed from the observations of polar-orbiting

instruments, including Aura OMI and SNPP OMPS, and it reveals a sizeable dependence of ice GLER values on the viewing-illumination geometry, indicating that the reflection from ice is significantly anisotropic. Since the much higher surface radiance $I_s$ (e.g., Fig. 6 blue line), the Lambertian treatment of ice surface can lead to large AMF errors. However, the ice GLER varies within a small range (0.94 to 0.98) and hence ice reflection has weak anisotropy for low SZA and VZA ( < 70°). Because the stronger $O_3$ absorption and Rayleigh scattering at shorter wavelengths reduce the fraction of direct solar beam but increase

that of the diffuse radiation reaching the surface, further weakening the BRDF effect, the error of Lambertian treatment of ice surface in the sampling of atmospheric $O_3$ absorption is suppressed for the low SZA and VZA observations.

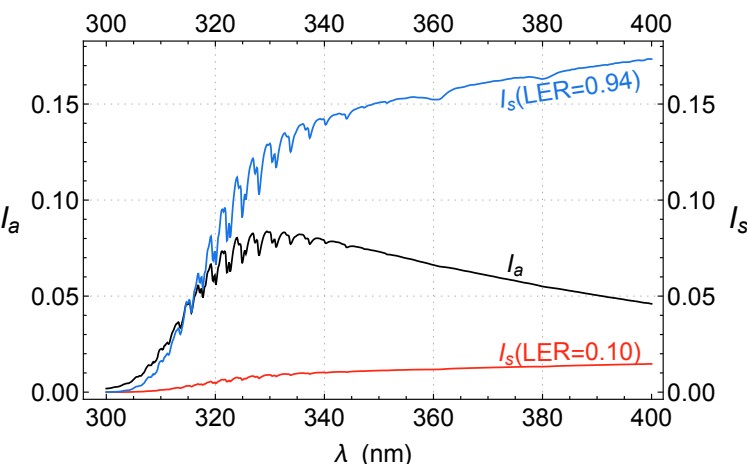

**Figure 6.** Example path radiance, $I_a$, and surface radiance $I_s$ for $\theta_s = 45°$, $\theta_v = 40°$, and $\phi = 135°$. $I_a$ is the middle line in black, and $I_s$ for LER = 0.1 and LER =0.94 are the lower (red) and upper (blue) lines, respectively.

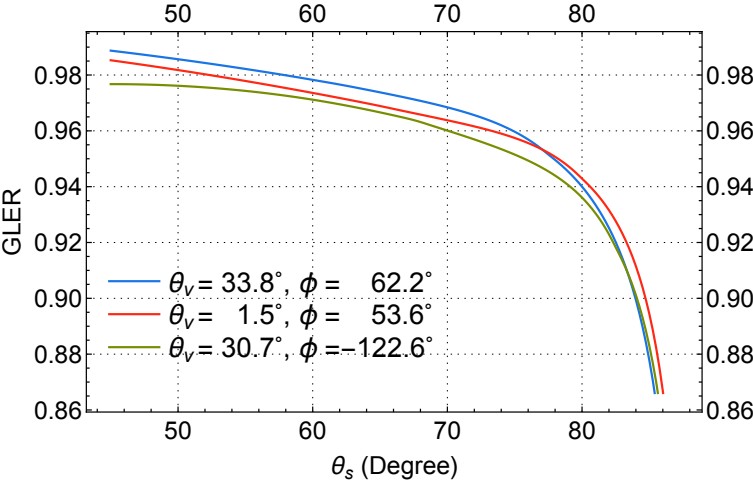

**Figure 7.** Climatological Antarctic GLER values at 331 nm as functions of SZA ($\theta_s$) for three viewing geometries, revealing a significant dependence of ice GLER on the viewing-illumination geometry.

## 2.3 Particle Scattering and Absorption

Atmospheric particles, including clouds and aerosols, reside mostly in the troposphere and cover a large portion ($\sim$67% by clouds alone, King et al. 2013) of the Earth's surface. Radiative transfer modeling of sunlight through a particle-laden atmosphere can be performed to quantify the TOA contributions from possible light paths, provided that the optical (scattering and absorption) properties of these particles, their amounts, and vertical distributions are specified. However, for UV remote sens-



ing observations, the quantitative information about particles needed for radiative transfer modeling is in general not known sufficiently, precluding their explicit treatment. In this section, we describe an implicit treatment of atmospheric particles for the simulation of measured radiances with the mean photon path approximately matching that through the particle-laden atmosphere.

Atmospheric particles scatter and possibly absorb UV photons, thus can significantly alter their paths through layers from closely above the particles down to the ground surface, usually shortening the path lengths below while lengthening those above the particles. Observing from space, the apparent effect of atmospheric particles is the enhancement of the TOA radiance contributed by backscattering from them. Since this effect is very similar to the consequence of an increased surface albedo, it is often referred to as the albedo effect. The albedo effect can be modeled by placing in a molecular atmosphere an elevated bright

surface that partially covers an IFOV. This treatment is called the mixed Lambertian-equivalent reflectivity (MLER) model, which is frequently employed by many algorithms for trace gas retrievals. Based on the MLER model, the TOA radiance for an IFOV is expressed as

$$I_{TOA} = I_g(R_g, p_g)(1 - f_c) + I_c(R_c, p_c)f_c, \tag{8}$$

the weighted sum of two independent contributions $I_g$ and $I_c$. Here $I_g$ is the radiance from the cloud-free portion of the IFOV,

containing a Lambertian surface of reflectivity $R_g$ at pressure $p_g$. Similarly, $I_c$ is from the cloudy portion, and $f_c$ is the cloud fraction and $R_c$ the reflectivity of the Lambertian surface at pressure $p_c$.

The MLER model can reproduce measured radiances $I_m$ through the determination of cloud fraction $f_c$. First, the scene LER $r_s$ at surface pressure $p_g$ is estimated using Eq. (6). If $r_s$ is less than or equal to the climatological LER value $R_g$ (e.g. Kleipool et al., 2008), this IFOV is treated as particle-free scene ($f_c = 0$). If $r_s$ is greater than or equal to the LER value for

cloud $R_c = 0.8$ (Koelemeijer and Stammes, 1999; Ahmad et al., 2004), this IFOV is treated as fully cloud covered ($f_c = 1$). When $r_s$ is in between $R_g$ and $R_c$, the cloud fraction is inverted from Eq. (8), which yields

$$f_c = \frac{I_M - I_g}{I_c - I_g}. \tag{9}$$

In case of $f_c = 0$ or 1, surface LER $r_g$ or cloud LER $r_c$ is determined using Eq. (6) to ensure that modeled radiance $I_{TOA}$ is equal to the measurement $I_M$. Figure 8 shows cloud fractions ($f_c$) as a function of wavelength for several examples of

particle-laden atmospheres.

The radiance intensity scattered from atmospheric particles varies with wavelength smoothly without high-frequency spectral structures. For instance, the contributions to TOA radiances ($I_{TOA}$) from backscattering by meteorological clouds change smoothly and slowly with wavelength (see Fig. 8, the CLD curve). The selection of $R_c = 0.8$ facilitates the MLER model to closely simulate the spectral variation of clouds observed from space (Ahmad et al., 2004), such that retrieved $f_c$ has a small

spectral variation (i.e., $f_c$ nearly the same for different wavelengths) for most cloudy observations. The small and smooth change of $f_c$ with wavelengths allows its extrapolation to provide a reliable estimate of $f_c$ at shorter wavelengths from those determined at longer wavelengths.

Certain types of aerosols, such as continental aerosols containing soot, smoke from fires, mineral dust from deserts, and ash from volcanic eruptions, both scatter and absorb UV photons passing through them. Usually, aerosol absorptions cause





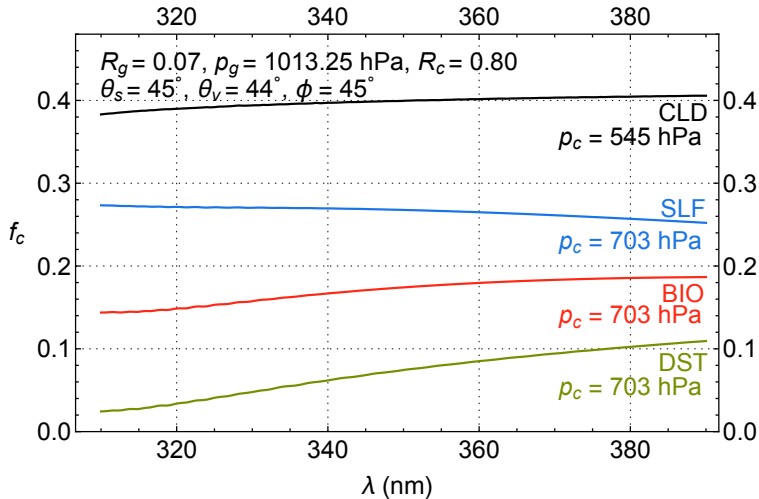

**Figure 8.** Four examples of cloud fractions ($f_c$) derived from explicitly modeled TOA radiances for particle-laden atmospheres. The first of these is the atmosphere with a 1.5-km-thick layer of C1 cloud (CLD, Deirmendjian 1969) with a single scattering albedo $\omega = 1$ and an optical thickness $\tau = 5$ at 340 nm centered at 5 km altitude (or pressure level of 545 hPa). The others are atmospheres with a 1-km-thick layer of aerosols, including SLF ($\omega = 0.996$), BIO ($\omega = 0.921$), and DST ($\omega = 0.900$) aerosols (SLF, BIO, and DST models are taken from Torres et al. 2007), with an optical thickness $\tau = 1.5$ at 340 nm centered at 3 km altitude (or pressure level of 703 hPa). The insets list the MLER parameters, $R_g$, $p_g$, $R_c$, and $p_c$, as well as the angles ($\theta_s$, $\theta_v$, and $\phi$) that specify the viewing-illumination geometry.

the underlying surface (including clouds) to appear darker, more so at shorter wavelengths. The aerosol extinction (i.e., the combination of absorption and scattering) impact to $I_{TOA}$ is a smooth function of wavelength (see Fig. 8, curves for weakly absorbing sulfate-based aerosols (SLF), carbonaceous aerosols from biomass burning (BIO), and mineral dust (DST)) (Torres et al., 2007). Therefore $f_c$ (when $f_c > 0$, from Eq. 9 ) or $r_g$ (when $f_c = 0$, from Eq. 6) determined at longer wavelengths where

atmospheric absorption is weak, maybe linearly extrapolated to $O_3$ sensitive wavelengths for estimation of contributions to TOA radiance from surface reflection and particle backscattering (referred to as the $r_g f_c$ extrapolation method hereafter).

     The UV aerosol index (AI) (Herman et al., 1997; Torres et al., 1998), which measures the deviation of spectral variation of TOA radiance from that of a pure molecular atmosphere, is proportional to the spectral slope $c_l$ used in the $r_g f_c$ extrapolation scheme. Algebraically, AI is calculated as the N-value (defined as $-100 \log_{10} I$) difference between the modeled ($I_{TOA}$) and

the measured ($I_M$) radiances at a wavelength $\lambda$

$$AI = 100 \, \log_{10} \frac{I_M(\lambda)}{I_{TOA}(\lambda, R_e)} \tag{10}$$

$$= 100 \, c_l \, \Delta\lambda \left. \frac{\partial \log_{10} I_{TOA}(\lambda, R)}{\partial R} \right|_{R=R_e}. \tag{11}$$

Here, the modeled radiance $I_{TOA}(\lambda, R_e)$ is calculated for a molecular atmosphere with an estimated reflectivity parameter $R_e$, which may be the LER value $r_s$ or the MLER cloud fraction $f_c$ determined at a well-separated wavelength ($\lambda + \Delta\lambda$). The pair of

wavelengths used for the AI calculation are in the UV spectral range with weak molecular absorption and their separation $\Delta\lambda$



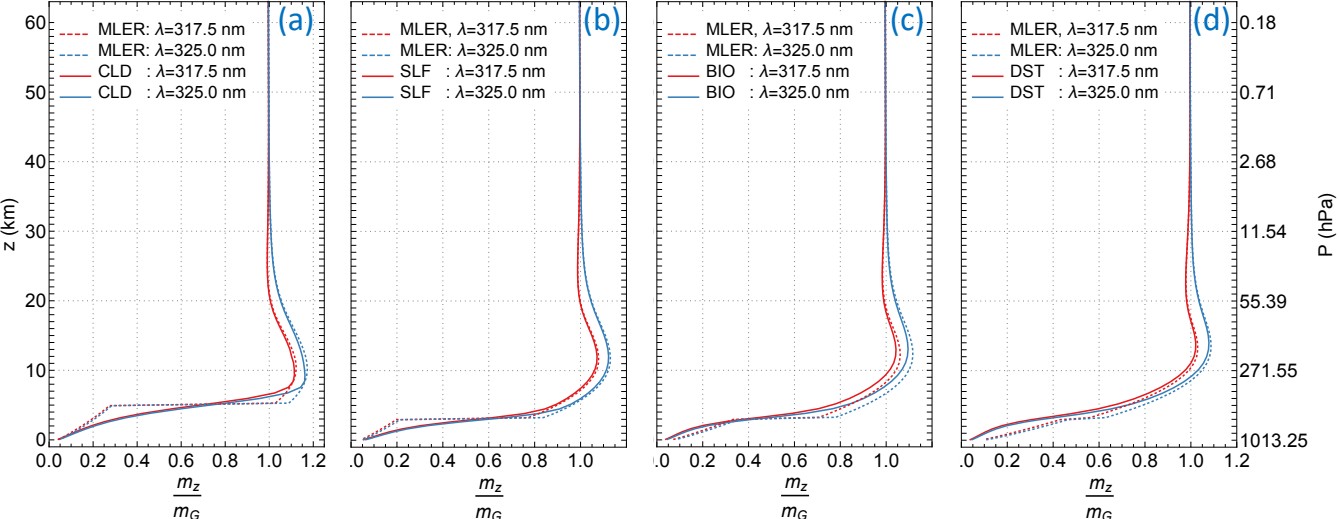

**Figure 9.** Mean photon path lengths $m_z$, normalized by the geometric AMF $m_G$, of EPIC bands 1 and 2 as functions of altitude $z$ for particle-laden atmospheres and their MLER treatments. See the caption of Fig. 8 for the description of aerosol characterizations, MLER treatments, and the viewing-illumination geometry.

should be sufficiently large ($> 10$ nm) to capture the spectral contrast of Rayleigh scattering. Using $I_M(\lambda) = I_{TOA}(\lambda, R_m) = I_{TOA}(\lambda, R_e + \Delta R)$, since the reflectivity parameter $R_m$ is derived from $I_M(\lambda)$ and $\Delta R = R_m - R_e = c_l \, \Delta\lambda$, we arrive at Eq. (11) from the definition of AI, Eq. (10). In short, the spectral slope $c_l$ is equivalent to the AI, which is significantly positive for particles (such as smoke, dust, and volcanic ash) with large absorption and slightly positive to negative for non-

absorbing and weakly absorbing particles (such as clouds and sulfate aerosols). Note that for the conventional AI (a.k.a. LER AI) calculation, radiance $I_{TOA}$ is modeled for a Rayleigh scattering-only atmosphere over a Lambertian surface. To capture the spectral slope of the $r_s f_c$ extrapolation scheme, we switch the LER treatment with the MLER modeling of $I_{TOA}$ for AI calculation. The resulting MLER AI is usually higher than the corresponding LER AI when $f_c > 0$, but otherwise can be similarly used to indicate the presence of UV-absorbing aerosol.

The MLER treatment enables the modeling of measured radiances without the knowledge of the optical properties or the full vertical distributions of atmospheric particles. The accuracy of the modeled radiances at the extrapolated wavelengths depends on how close the MLER parameter ($r_g$ or $f_c$) follows the linear relationship among different wavelengths. In reality, the spectral dependence of natural surface reflection ($r_g$) or particle scattering and absorption ($f_c$) are nonlinear, though moderately as exemplified in Figs. 4b and 8, therefore $r_g f_c$ extrapolation yields small errors in $r_g$ or $f_c$ at the extrapolated wavelengths.

The radiance uncertainties associated with the $r_g f_c$ extrapolation error are below 1% for the vast majority of remote sensing observations. Higher radiance uncertainties usually occur in the presence of highly elevated or strongly absorbing aerosols. These observations may be flagged with high AI values.



In addition to the mostly small radiance errors at the extrapolated wavelengths, the MLER treatment can simulate the photon sampling of particle-laden atmospheres with a diverse range of particle types and vertical distributions. Figure 9 shows comparisons of mean photon path lengths of particle-laden atmospheres with those from the corresponding MLER treatments. These comparisons illustrate that the layer mean photon paths based on the MLER model deviate from those of the particle-

laden atmospheres, mostly in the region immediately above the particles down to the underlying surface. These deviations diminish with higher altitudes where lower air density reduces the chance of photons being scattered. Since the vast majority of clouds and aerosols are in the lower troposphere ($<\sim10$ km), the MLER treatment does not introduce significant AMF errors in accounting for $O_3$ absorption, which occurs mostly in the stratosphere. This is similar to how the Lambertian treatment of surface reflection works for the estimation of total $O_3$ absorption (see section 2.2).

The MLER treatment relies on a few adjustable parameters, including the cloud fraction $f_c$ and cloud pressure $p_c$, to model a vast range of conditions encountered in remote sensing of Earth's atmosphere. The cloud fraction $f_c$, obtained directly from radiance measurements using Eq. (9), provides an estimate of the cloud amount in an IFOV. The pressure $p_c$ of the elevated Lambertian surface needs to be set at a proper level to best approximate the layer mean photon paths of a particle-laden atmosphere. As seen in Fig. 9, the optimal placement of the elevated Lambertian surface is within the particle layer, as $p_c$

locates too high or too low from the optical centroid pressure (OCP) (Joiner and Vasilkov, 2006; Vasilkov et al., 2008) would make layer mean photon paths deviate further from those of the particle-laden atmosphere. The effective cloud pressures retrieved from the EPIC measurements of $O_2$ A-band (Y.Yang et al. 2019) are usually located within the particle vertical distributions and therefore used to set the cloud pressures $p_c$ for processing EPIC observations.

The use of OCP for $p_c$ enables the MLER model to account for the measurement sensitivity change when a layer of particles

is introduced into the atmosphere: enhancing the photon attenuation by absorbers inside and above the layer, while reducing them below, as the mean photon paths or AMFs from the MLER model lengthen above $p_c$, but shorten below it, as illustrated in Fig. 9. Since the MLER model captures the enhancement and shielding effects on trace gas absorption by atmospheric particles, it is widely adopted due to its simplicity for retrievals of trace gases besides $O_3$, such as $NO_2$ and $SO_2$ in the troposphere. However, sizeable AMF errors are prevalent for modeling tropospheric absorptions based on the MLER treatment, which

usually yields significantly different mean photon paths from those of explicit treatment in the troposphere.

### 2.4 Inelastic Molecular Scattering

The scattering of sunlight with atmospheric constituents is mostly elastic, i.e., the energy and thus the wavelength of a photon remain the same before and after the interaction. But a small portion ($\sim4\%$) of molecular scattering is inelastic, resulting in energy gain or loss of the scattered photons. Specifically, the rotational Raman scattering (RRS) from air molecules (such as

nitrogen and oxygen) can alter the wavelengths of scattered photons, with UV wavelength shifts $\Delta\lambda < \pm 2$ nm (Joiner et al., 1995; Chance and Spurr, 1997; Vountas et al., 1998). These inelastic scatterings cause the filling-in of telluric lines (i.e., trace gas absorption features) and solar Fraunhofer lines (also known as the Ring effect, which was first noticed by Grainger and Ring 1962).



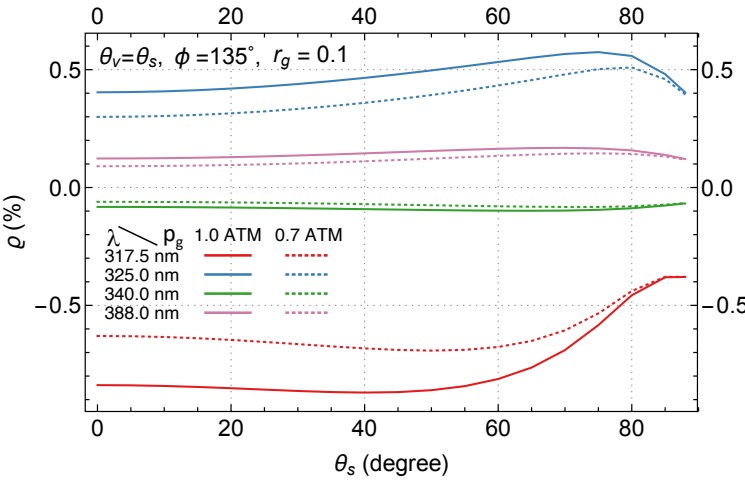

**Figure 10.** Filling-in factors ($\varrho$) for EPIC UV bands as a function zenith angle at two surface pressures $p_g = 0.7$ ATM and $p_g = 1.0$ ATM, with a surface albedo of $r_g = 0.1$.

The filling-in effect is a function of wavelength and depends on the optical properties of the atmosphere, the viewing and illumination geometry, and the surface reflectivity and pressure. The filling-in effect also depends on the ISRF, especially on the instrument spectral resolution, which is the width of its ISRF, since the measured radiance of a band is a convolution of spectral radiance and the ISRF (see Eq. 1). This effect is quantified with the filling-in factors, defined as $\varrho = (I_{RRS} - I_{ELA})/I_{ELA}$,

where $I_{ELA}$ is the TOA radiance calculated assuming all molecular scattering is elastic, while $I_{RRS}$ includes the inelastic (RRS) contributions. To illustrate the significance of RRS, we show in Fig. 10 examples of the filling-in factors, calculated for EPIC bands using the scalar LIDORT-RRS radiative transfer code (Spurr et al., 2008). Since RRS is weakly dependent on polarization, a scalar radiative transfer model, from which both $I_{ELA}$ and $I_{RRS}$ are calculated without including radiation polarization, can accurately provide filling-in factors (Landgraf et al., 2004; Wagner et al., 2010).

The filling-in factors provide estimates of the modeling errors in $I_{TOA}$ when RRS contributions are neglected, and results in Fig. 10 show variations of modeling errors with different observing conditions. These errors are usually systematic for a spectral band and are between half to one percent for measurements of EPIC bands 1 and 2. These errors are sufficiently large that corrections are required for achieving high ($\sim$1%) $O_3$ retrieval accuracy. The filling-in factors ($\varrho$), modeled using a scaler code (like LIDORT-RRS), may be used to correct the results ($I_{TOA}$) from vector radiative transfer codes (e.g. Dave, 1964;

Spurr, 2006) that perform elastic modeling only, i.e., the RRS corrected TOA radiance $= I_{TOA}(\varrho+1)$.

## 3  $O_3$ and Temperature Vertical Profiles

As shown in section 2.1, the $O_3$ vertical distribution or profile directly affects the magnitude of a measured radiance in the spectral region with significant $O_3$ absorption. Hence the interpretation of radiance change due to $O_3$ absorption requires some knowledge of its profile. In general, the retrieval of quantitative information about a gaseous absorber (such as $O_3$ and $SO_2$)





requires a model to prescribe its vertical distribution. The skill of this model in representing the actual vertical distribution of the absorber contributes significantly to the quantification accuracy. In this section, we describe a recently developed $O_3$ profile model for remote sensing retrieval algorithms and its improvements over the model commonly used by other total O3 algorithms.

Ozone is naturally present throughout the atmosphere and its spatial and temporal distribution controlled by atmospheric processes of $O_3$ production, destruction, and transport. The $O_3$ distribution exhibits a high abundance of $O_3$ in the stratosphere and a minor portion ($\sim$10%) in the troposphere, with the peak $O_3$ concentration occurring at a lower altitude as the latitude increases towards the poles. These characteristics are well captured by $O_3$ profile climatologies (e.g. Fortuin and Kelder, 1998; McPeters et al., 2007; McPeters and Labow, 2012), which provide the mean and variance of $O_3$ vertical distribution as a

function of latitude and calendar month. These climatologies also reveal that $O_3$ profile has the highest variability in the upper troposphere and lower stratosphere (UTLS), contributing the most to the natural variations in total $O_3$. This high $O_3$ variability is the consequence of atmospheric movements that blend air masses with different $O_3$ concentrations, such as uplifting of $O_3$ poor air in the troposphere or lowering of $O_3$ rich air in the stratosphere resulting from the rise and fall of the tropopause. Predictors of $O_3$ profile shape, including tropopause pressure and total $O_3$ columns, are developed to capture the dynamical

influences on $O_3$ vertical distributions, resulting in the construction of tropopause-sensitive (Wei et al., 2010; Bak et al., 2013; Sofieva et al., 2014) and total-column-dependent (Wellemeyer et al., 1997; Bhartia and Wellemeyer, 2002; Lamsal et al., 2004; Labow et al., 2015) $O_3$ profile climatologies.

The $O_3$ profile model for the Total Ozone Mapping Spectrometer Version 8 (TOMS-V8) total $O_3$ algorithm combines the latitude-dependent monthly mean Labow-Logan-McPeters (LLM) climatology (McPeters et al., 2007) with the latitude- and

total-column-dependent annual mean climatology (Bhartia and Wellemeyer, 2002) to determine the $O_3$ profile as a function of latitude, time (day of year, DOY), and total $O_3$ column. This model has been adopted by nearly all the contemporary total $O_3$ algorithms (e.g. Bhartia and Wellemeyer, 2002; Eskes et al., 2005; Veefkind et al., 2006; Van Roozendael et al., 2006; Lerot et al., 2010; Loyola et al., 2011; Van Roozendael et al., 2012; Lerot et al., 2014; Wassmann et al., 2015), owing to its capability of characterizing $O_3$ profile variation with the total column.

To improve the representation of $O_3$ profile, we construct both tropopause-dependent and total-column-dependent climatologies using the Modern-Era Retrospective Analysis for Research and Applications version 2 (MERRA-2, Bosilovich et al. 2015; Gelaro et al. 2017) $O_3$ record between 2005 and 2016. The total-column-dependent climatology, named M2TCO3, is more appropriate for use as the $O_3$ profile model needed by a total $O_3$ algorithm, as it is generally more reliable than the tropopause-dependent version in prescribing realistic $O_3$ profiles (Yang and Liu, 2019).

Figure 11 compares daytime M2TCO3 (Yang and Liu 2019, referred to as M2TCO3 hereafter) and TOMS-V8 profiles for two months and four latitude zones, illustrating the similarities and differences between the two $O_3$ models. Both show north-south asymmetry, i.e., profiles in the northern hemisphere differ from those in the southern hemisphere for the corresponding months and latitude zones (e.g., September and 60°S–50°S vs. March and 50°N–60°N in Fig. 11), substantial seasonal variations (e.g., 60°S–50°S, March vs. September in Fig. 11), strong dependence on latitude, exhibiting lower altitudes of $O_3$

concentration peaks at higher latitudes for similar total columns, and characteristic dependence on the total column, which gets





**Figure 11.** Profile comparisons between M2TCO3 and TOMS-V8 for two months (March and September) and four latitude zones: 60°S–50°S, 30°S–20°S, 20°N–30°N, and 50°N–60°N. Colored solid lines represent M2TCO3 profiles, while the dotted ones for TOMS-V8 profiles. The color of a solid line indicates the percentage occurrence of the climatological profile, and its line legend displays the mean tropopause altitude and the mean total column $O_3$ of the profile. The solid black lines represent the downgraded M2TCO3 (i.e., the monthly zonal mean) profiles and dotted lines are TOMS-V8 monthly zonal mean (i.e., the LLM climatological) profiles. Here pressure altitude is defined as $Z^* = 16 \log_{10}\left[\frac{p_s}{p}\right]$, where $p$ is pressure level (in hPa) and $p_s = 1013.25$ hPa.

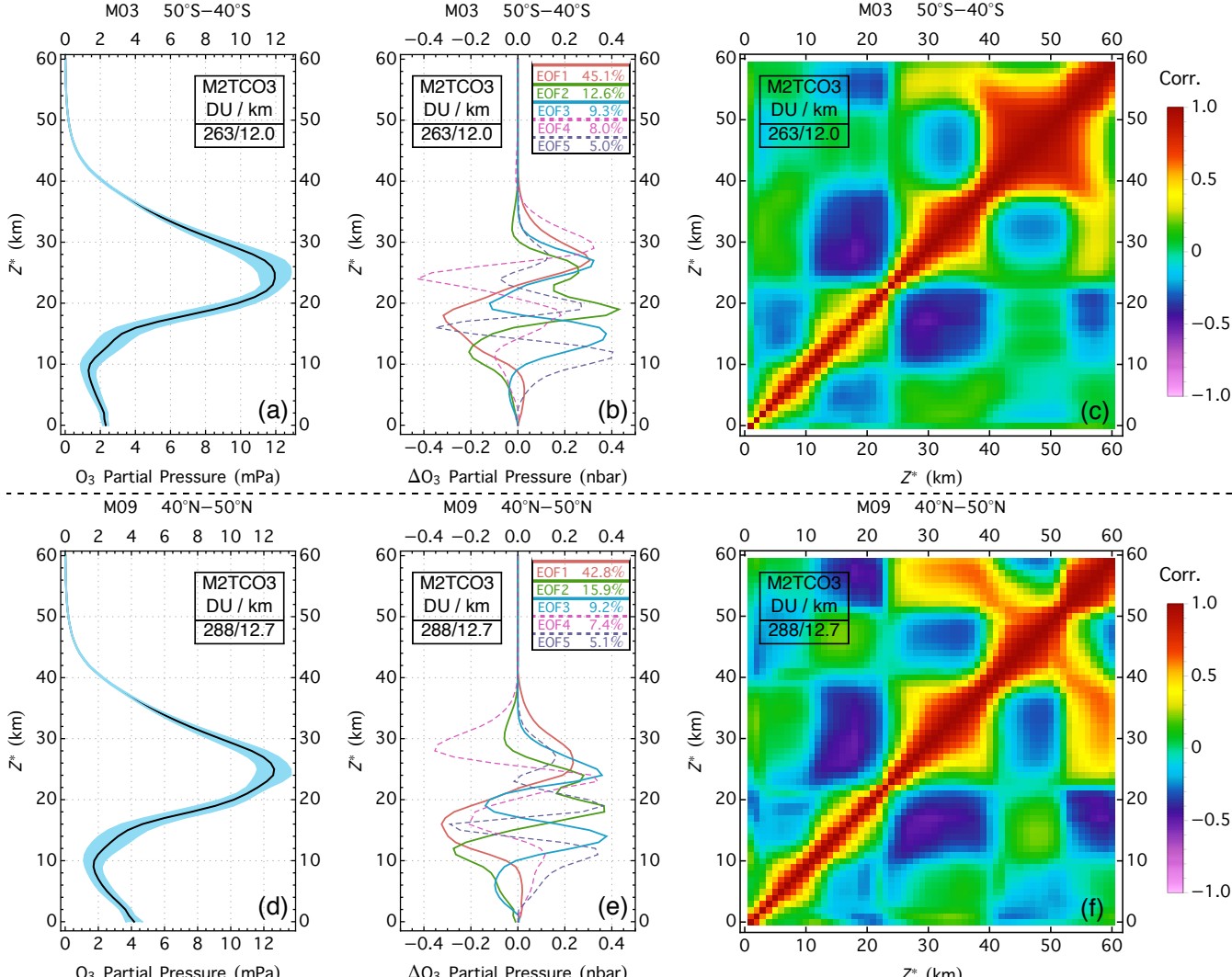

**Figure 12.** Examples of M2TCO3 climatological profiles for the southern midlatitude zone in March (panel a) and the northern midlatitude zone in September (panel b), the associated correlation matrices (panels c and f), and the corresponding modal $O_3$ profiles (panels b and e). The blue shaded areas in panels a and d are within one standard deviation of the mean. The correlation matrices in panels c and f are standardized (i.e., diagonal element normalized to 1) covariance matrices. The five modal profiles in panels b and e are the first five ordered eigenvectors (also known as empirical orthogonal functions or EOFs) of the corresponding covariance matrices, with percentages of the profile variance explained by the EOFs displayed in the line legends. The text box in each panel displays the average tropopause altitude (in km) and the average total $O_3$ column (in DU) for the climatological profile.

smaller with a higher $O_3$ peak altitude (e.g., March and 50°N–60°N in Fig. 11). Figure 11 shows good agreements of zonal mean profiles (e.g., close matches between solid black and dotted black curves in each panel of this figure), but significant dif-



ferences between M2TCO3 and TOMS-V8 profiles for similar total columns. These differences are due to TOMS-V8's use of annual mean column-dependent climatology to account for profile variations with the total column throughout the year (Bhartia and Wellemeyer, 2002), thus ignoring the significant seasonal dependence. An additional deficiency of TOMS-V8 contributing to the differences is its inadequate representation of latitude-dependent $O_3$ profile variation with the total column, including

broad ($30°$) latitude zones and omission of north-south asymmetry. These deficiencies are eliminated with M2TCO3, which improves the realism of $O_3$ profile representation.

In short, M2TCO3 better captures the dynamical changes and spatiotemporal variations in $O_3$ profiles with higher resolutions in total $O_3$ column, latitude, and time. Taking into account the substantial change of atmospheric $O_3$ over a long time, M2TCO3 is more accurate to represent atmospheric $O_3$ vertical distribution from recent past to near future than the TOMS-V8 model,

which was compiled from earlier satellite and ozonesonde data (mostly from the 1980s and 1990s, Wellemeyer et al. 1997; McPeters et al. 2007). Hence we use the M2TCO3 climatology as the $O_3$ profile model for total $O_3$ retrieval from EPIC.

The M2TCO3 climatology contains not only mean profiles that represent the likely $O_3$ vertical distributions, but also the modal $O_3$ adjustment profiles that specify the probable deviations from the means. These modal profiles are determined from the $O_3$ profile covariance statistics, as illustrated in Fig. 12, showing examples of M2TCO3 climatological $O_3$ profiles and

the associated modal profiles, which are the eigenvectors (also known as the Empirical Orthogonal Functions or EOFs) of the profile covariance matrices. Algebraically the representation of an $O_3$ profile $\mathbf{X}$ is expressed as

$$\mathbf{X} = \mathbf{X_m}(\mathbf{v}) + \sum_{k=1}^{p} \gamma_k \, \mathbf{e}_k(\mathbf{v}), \tag{12}$$

where $\mathbf{X_m}(\mathbf{v})$ is a climatological profile that depends on a set of variables $\mathbf{v}$, which for M2TCO3 consists of the total column ($\Omega_0$), time, and location. $\mathbf{e}_k(\mathbf{v})$ is the $k^{\text{th}}$ modal profile, $\gamma_k$ the $k^{\text{th}}$ coefficient, and $n$ the number of $\mathbf{e}_k(\mathbf{v})$, with a maximum

equal to the number of levels used to represent an $O_3$ profile in the climatology. Usually, a few modal profiles are sufficient to account for the majority of profile variance. For example in Fig. 12, the first five EOFs (panels b and e) of the covariance matrices (panels c and f) account for 80% profile variances (blue shaded area in panels a and d). An actual $O_3$ profile $\mathbf{X}$, which deviates invariably from the mean $\mathbf{X_m}$, can be accurately represented using Eq. (12) with a small number of expansion coefficients $\omega_k$. Much like the mean the profile $\mathbf{X_m}$ represents the most probable vertical distribution of $O_3$, the modal profiles,

$\{\mathbf{e}_k, k = 1 \ldots\}$, describe the most, the second most, and so on, likely vertical patterns of deviations from the mean profile. Each modal profile describes a rearrangement, like shifting, shrinking, or broadening, of the mean profile without substantially changing the total column. With these modal profiles constraining how a profile can be adjusted, the retrieval algorithm can exploit the $O_3$ profile information contained in multi-spectral measurements to improve the $O_3$ profile representation by determining one or more linear expansion coefficients $\{\gamma_k, k = 1 \ldots\}$. Note that for most total ozone algorithms, the $O_3$ profile

representation is limited to the climatological mean only, equivalent to restricting $\omega_k = 0$ for all $k$ in Eq. (12).

The total column is a good predictor of an $O_3$ profile, especially accurate for the shape in the stratosphere, but less so in the troposphere. Tropospheric $O_3$ exhibits characteristic spatiotemporal distribution, which is captured in the MERRA-2 tropospheric $O_3$ climatolgoy (Yang and Liu, 2019). To better represent the $O_3$ profile, the tropospheric part of a column-dependent M2TCO3 profile, $\mathbf{X_m}$, is scaled with the ratio of the MERRA-2 climatological tropospheric column to the tropospheric col-





umn integrated from the downgraded M2TCO3 profile (see Fig. 11 for sample M2TCO3 and downgraded M2TCO3 profiles). In other words, the profile $\mathbf{X_m}$ in Eq. (12) has its tropospheric part tied to the spatiotemporally varying climatological tropospheric column, to which the tropospheric column of the mean $\mathbf{X_m}$ profile (obtained by averaging over the different column amounts) is matched.

In addition to knowledge of profiles of light-absorbing trace gases, such as $O_3$ and $SO_2$, radiative transfer modeling of measured radiance requires knowledge of the atmospheric temperature profile because the absorption cross-sections of these trace gases depend on temperature significantly. For total $O_3$ retrieval from EPIC, this knowledge is taken from the temperature profile climatology created from MERRA-2 data together with the ozone profile climatlogy (Yang and Liu, 2019). This temperature climatology provides mean temperature profiles corresponding to the climatological $O_3$ profiles, capturing the
dependence of temperature profile on season and location, as well as the variation of temperature with $O_3$ profile. It is an improvement over the TOMS-V8 temperature profile climatology, which provides latitude-month dependent temperature profiles, but without accounting for the strong correlation between temperature and $O_3$ profiles.

## 4   Inversion Technique

Section 2 describes algorithm physics treatments of interactions of solar radiation with atmospheric particles and surfaces to
enable RT modeling of photons traversing through a molecular atmosphere to reproduce the measured TOA radiances with photons that follow the paths similar to those through the actual atmosphere and therefore establish the relationship between spectral measurements and the atmospheric state, as well as surface reflectivity and instrumental parameters. At its core, the RT modeling sets up a forward mapping from the vertical distributions of gaseous absorbers and the surface reflectivity parameters to measured TOA radiances. The retrieval of gas absorbers, such as $O_3$ and $SO_2$, is the inverse of this mapping, i.e., to find
their vertical distributions and the surface reflectivity parameters for which forward modeling closely reproduces the measured TOA radiances. However, this inverse mapping is inherently an ill-posed problem, as the solution is not unique, i.e., more than one set of profiles and surface parameters can yield the same measurements. This problem is made worse with measurement uncertainties, which expand the profile and surface combinations that can reproduce, within error bars, the measured spectra.

For successful inversion, analytical constraints are placed on the profiles of gas absorbers and the spectral variations of
ground reflectivity and atmospheric particle (aerosol and cloud) back-scattering. For $O_3$ retrieval, Eq. (12) embodies the profile constraint, while the MLER model with $r_s f_c$ extrapolation regulates the surface reflection and particle back-scattering. These constraints control the the dimension of the inverse mapping space and manifest themselves as the retrieval (i.e., adjustable) parameters, which, in the case of $O_3$ retrieval, consist of total $O_3$ column $\Omega$, a number ($p$) of modal expansion coefficients $\{\gamma_k, k = 1 \dots p\}$, surface LER ($r_s$) or cloud fraction ($f_c$), and a number ($q$) of polynomial coefficients $\{c_l, l = 1 \dots q\}$ of the
$r_s f_c$ extrapolation. The set of adjustable parameters forms the state vector ($\mathbf{x}$) whose length ($n$) is the dimension of inverse mapping space. Proper selection of adjustable parameters by limiting the number of the modal coefficients ($p \geq 0$) and the polynomial coefficients ($q \leq 1$) ensures the inverse problem is well-posed and simultaneously maximizes the amount of infor-





mation collected from the spectral measurements. Here $p = 0$ indicates no modal expansion, equivalent to restricting the profile to a climatological column-dependent $O_3$ profile, and $q = 0$ for the spectral invariant reflectivity parameter.

## 4.1 Exact Solution

Conceptually, the inversion is to find the state vector ($\mathbf{x}$) that satisfies a set of $m$ simultaneous equations, $\{\Delta y_i = 0, i = 1 \ldots m\}$, one for each spectral band difference, $\Delta y_i = \ln I_M(\lambda_i) - \ln I_{TOA}(\mathbf{x}, \lambda_i)$, between the radiance measurement $I_M$ and the forward modeling $I_{TOA}$. Here $\lambda_i$ the wavelength that characterizes the $i^{th}$ ($1 \leq i \leq m$) spectral band and $\Delta y_i$ the residual of this band. In matrix form, the $m$ simultaneous equations can be expressed as

$$\Delta \mathbf{y} = 0, \tag{13}$$

where $\Delta \mathbf{y}$ is residual column vector $\{\Delta y_i, i = 1 \ldots m\}$. Since the forward mapping $I_{TOA}(\mathbf{x})$ is a nonlinear function of the state vector $\mathbf{x}$ and has no analytical inverse, the solution to Eq. (13) is usually obtained iteratively. The iteration is started with an initial (i.e., iteration number $L = 0$) state vector $\mathbf{x}_L$ to linearize the equation between residuals and the state vector

$$\Delta y_i = \ln I_M(\lambda_i) - \ln I_{TOA}(\mathbf{x}_L, \lambda_i) - \sum_{j=1}^{n} (x_j - x_{Lj}) \frac{\partial \ln I_{TOA}(\mathbf{x}, \lambda_i)}{\partial x_j} \bigg|_{\mathbf{x}=\mathbf{x}_L}, \tag{14}$$

where $x_j$ and $x_{Lj}$ are the $j^{th}$ components of $\mathbf{x}$ and $\mathbf{x}_L$ respectively, $\Delta x_j = x_j - x_{Lj}$ the $j^{th}$ components of state adjustment vector, and $K_{ij} = \frac{\partial \ln I_{TOA}(\mathbf{x}, \lambda_i)}{\partial x_j} \big|_{\mathbf{x}=\mathbf{x}_L}$ the Jacobian, also known as the weighting function for the retrieval parameter $x_j$ at spectral band $\lambda_i$. The $m$ residual elements, each written in Eq. (14), can be expressed in matrix form as

$$\Delta \mathbf{y} = \Delta \mathbf{y}_L - \mathbf{K} \Delta \mathbf{x}, \tag{15}$$

where $\Delta \mathbf{y}_L$ is the column vector $\{\ln I_M(\lambda_i) - \ln I_{TOA}(\mathbf{x}_L, \lambda_i), i = 1 \ldots m\}$, $\Delta \mathbf{x} = \mathbf{x} - \mathbf{x}_L$ the state adjustment vector, and $\mathbf{K}$ the $m \times n$ Jacobian matrix with the $\{K_{ij}, i = 1 \ldots m, j = 1 \ldots n\}$ as its elements. Putting Eq. (15) into Eq. (13) yields

$$\Delta \mathbf{y}_L = \mathbf{K} \Delta \mathbf{x}, \tag{16}$$

which may be solved exactly (under strict conditions) to determine state adjustment vector $\Delta \mathbf{x}$. After each iteration, the linearization state vector is updated to

$$\mathbf{x}_{L+1} = \mathbf{x}_L + \Delta \mathbf{x}. \tag{17}$$

The final state $\mathbf{x}$ is found when the iteration converges, i.e., when the absolute change of state vector $\Delta \mathbf{x}$ is below a threshold.

The linear equation (Eq. 16) may be solved exactly only when the number of measurements is equal to the number of retrieval parameters (i.e. $m = n$) and the Jacobian matrix $\mathbf{K}$ is invertible (i.e., non-singular matrix), as exemplified in the well-known TOMS-V8 total $O_3$ algorithm (Bhartia and Wellemeyer, 2002). The TOMS-V8 algorithm determines the two-component state vector, $\mathbf{x} = \{\Omega, r_s \text{ or } f_c\}$, from radiance measurements of two spectral bands: one with low $O_3$ sensitivity to estimate the MLER parameter ($r_s$ or $f_c$), and the other with high $O_3$ sensitivity to derive total $O_3$ column $\Omega$. However, few other algorithms



adopt this inversion method, since it requires $m = n$ and $\mathbf{K}$ being a nonsingular matrix. Even if both these conditions are met, inverting Eq. (16) to obtain exact solutions tends to enhance the impact of measurement uncertainties (noises) on the retrieved results, as in cases that $\mathbf{K}$ matrices are nearly but not quite singular. These cases occur when the spectral variation of a Jacobian has some similarity or a high degree of correlation with that of another retrieval parameter, leading to algorithm difficulty in

distinguishing two retrieval parameters corresponding to the two Jacobians, thus yielding unstable retrieval results, such as in the case of simultaneous retrieval of total $O_3$ and $SO_2$ columns from EPIC UV measurements.

## 4.2   Direct Fitting

Since spectral measurements have errors and $m \neq n$ in general, the inversion is achieved by finding the solution $\mathbf{x}$ that minimizes the cost function

$$\Upsilon(\mathbf{x}) = \left\| \mathbf{S}_\epsilon^{-\frac{1}{2}} \Delta \mathbf{y} \right\|_2^2 = \Delta \mathbf{y}^T \mathbf{S}_\epsilon^{-1} \Delta \mathbf{y} \tag{18}$$

$$= \sum_{i=1}^{m} \left( \frac{\Delta y_i}{\mu_i} \right)^2, \tag{19}$$

where $\mathbf{S}_\epsilon$ is the measurement error covariance matrix, with its $i^{th}$ diagonal element equal to $\mu_i^2$. Here $\mu_i$ is the fractional standard deviation of radiance error of the $i^{th}$ band. In case of independent measurement error, i.e., no error correlation between different spectral bands, Eq. (18) can then be explicitly written as Eq. (19), which is the formulation of the least-squares method.

The minimization of the cost function $\Upsilon$ can be started by linearizing the residuals with an initial (i.e., iteration number $L = 0$) state vector $\mathbf{x}_L$. Substituting $\Delta \mathbf{y}$ (Eq. 15) into Eq. (18), we minimize this cost function to obtain the the state adjustment vector

$$\Delta \mathbf{x} = (\mathbf{K}^T \mathbf{S}_\epsilon^{-1} \mathbf{K})^{-1} \mathbf{K}^T \mathbf{S}_\epsilon^{-1} \Delta \mathbf{y}_L = \mathbf{G}_{\mathrm{DF}} \Delta \mathbf{y}_L, \tag{20}$$

which is the solution of linear weighted least-square regression. Here, $\mathbf{G}_{\mathrm{DF}} = (\mathbf{K}^T \mathbf{S}_\epsilon^{-1} \mathbf{K})^{-1} \mathbf{K}^T \mathbf{S}_\epsilon^{-1}$ is the direct fitting (DF)
gain matrix.

This procedure of iterative minimization of the difference between measurements and modelings to determine the bulk parameters is called the direct vertical column fitting (DVCF) algorithm. The DVCF algorithm is quite general and valid for both discrete-wavelength and hyperspectral measurements, as well as for as applicable for different types of retrieval parameters, such as MLER parameters, layer partial columns of various absorbing trace gases, and their total vertical columns.
This algorithm has been applied to retrievals of total $O_3$ vertical column (Joiner and Bhartia, 1997; Yang et al., 2004), combo of total $O_3$ and $SO_2$ vertical columns (Yang et al., 2007, 2009a, 2013), combo of $O_3$ and altitude-resolved $SO_2$ vertical columns (Yang et al., 2009b, 2010), and stratospheric and tropospheric $NO_2$ vertical columns (Yang et al., 2014). This algorithm is named DVCF to contrast with the DOAS (the Differential Optical Absorption Spectroscopy) method (Platt, 2017), which derives a slant column and then uses an air mass factor (AMF) at a single wavelength ($\lambda_0$) to converts it to a vertical column.
In general, the DVCF algorithm works well when the changes in radiance measurements responding to changes in the state vectors are significantly different between any two retrieval parameters, i.e., that columns of $\mathbf{K}$, which are the Jacobians





of a retrieval parameter at different wavelengths, exhibit significantly different spectral dependence from one another. This is usually true for any two bulk retrieval parameters over a sufficiently broad spectral range, such as total $O_3$ column ($\Omega$) and an expansion coefficient ($\omega_k$) of differential profile $\mathbf{e}_k$ (see Eq. 12), or the $SO_2$ vertical column and its layer altitude. With measurements from a broad spectral range, the DVCF algorithm can discriminate subtle spectral features contained in

hyperspectral measurements to enhance the retrieval accuracy (e.g., Yang et al. 2009b, 2010). Besides contrasting with the DOAS method, the name DVCF emphasizes vertical column because this algorithm is usually not suitable for traditional profile retrieval, due to the high similarity of partial column Jacobians between adjacent layers and hence the difficulty in distinguishing their partial columns.

### 4.3   Optimal Estimation

In many cases, such as sparse spectral sampling or narrow spectral range, the performance of the direct fitting inversion method may decline as a result of limited information contained in the spectral measurements. For stabilizing the retrieved results, the inversion process can be regulated with an additional constraint, which is frequently based on the *a priori* knowledge of the retrieval parameters. Algebraically, adding an *a priori* constraint to Eq. (18) yields a new cost function

$$\Upsilon(\mathbf{x}) = \Delta\mathbf{y}^T\mathbf{S}_\epsilon^{-1}\Delta\mathbf{y} + (\mathbf{x}-\mathbf{x}_a)^T\mathbf{S}_a^{-1}(\mathbf{x}-\mathbf{x}_a), \tag{21}$$

where $\mathbf{x}_a$ is the *a priori* state vector and $\mathbf{S}_a$ the *a priori* state vector covariance matrix. The first term on the right-hand side (r.h.s) of Eq. (21) strives to diminish the difference between measured and modeled radiances, performing the same function as the direct fitting retrieval, while the second r.h.s term seeks to reduce the deviation of retrieved $\mathbf{x}$ from the *a priori* $\mathbf{x}_a$. This *a priori* constraint effectively stabilizes the retrieval by guiding the state vector adjustment when the measurements contain little information to differentiate the contributions from different components of the state vector. Using the optimal estimation (OE)

technique (Rodgers, 2000) to minimize the cost function Eq. (21) yields a posterior state adjustment vector at the $L^{th}$ iteration

$$\Delta\mathbf{x} = (\mathbf{S}_a^{-1} + \mathbf{K}^T\mathbf{S}_\epsilon^{-1}\mathbf{K})^{-1}(\mathbf{K}^T\mathbf{S}_\epsilon^{-1}\Delta\mathbf{y}_L + \mathbf{S}_a^{-1}\Delta\mathbf{x}_{aL}) \tag{22}$$

$$= (\mathbf{S}_a^{-1} + \mathbf{K}^T\mathbf{S}_\epsilon^{-1}\mathbf{K})^{-1}\left(\mathbf{K}^T\mathbf{S}_\epsilon^{-1}\mathbf{K}\,\mathbf{G}_{\mathrm{DF}}\Delta\mathbf{y}_L + \mathbf{S}_a^{-1}\Delta\mathbf{x}_{aL}\right) \tag{23}$$

$$= \Delta\mathbf{x}_{aL} + \mathbf{S}_a\mathbf{K}_T(\mathbf{K}\mathbf{S}_a\mathbf{K}_T + \mathbf{S}_\epsilon)^{-1}(\Delta\mathbf{y}_L - \mathbf{K}\Delta\mathbf{x}_{aL}), \tag{24}$$

where $\Delta\mathbf{x}_{aL} = \mathbf{x}_a - \mathbf{x}_L$ and the primed quantities are defined in section 4.2. Inserting $\mathbf{I}_n = (\mathbf{K}^T\mathbf{S}_\epsilon^{-1}\mathbf{K})(\mathbf{K}^T\mathbf{S}_\epsilon^{-1}\mathbf{K})^{-1}$, an

$n \times n$ identity matrix in front of the term $\mathbf{K}^T\mathbf{S}_\epsilon^{-1}\Delta\mathbf{y}_L$ in Eq. (22) yields Eq. (23). At iteration $L = 0$, a state vector close to the actual one is sought to be the initial state vector $\mathbf{x}_0$, and a frequent selection is the *a priori* state vector: $\mathbf{x}_0 = \mathbf{x}_a$. This is a more robust inversion scheme that works for $m > n$, $m = n$, and $m < n$. Eq. (24) is often used in the case of $m < n$, as the inversion deals with an $m \times m$ (i.e., a smaller) matrix.

Eq. (23) describes the difference between the current and previous state vectors $\Delta\mathbf{x}$ as a combination of the direct fitting

solution $\mathbf{G}_{\mathrm{DF}}\Delta\mathbf{y}_L$ (see Eq. 20, which is derived without any *a priori* constraint) and the difference between the *a priori* and the previous state vectors $\Delta\mathbf{x}_{aL}$ weighted by matrices $\mathbf{K}^T\mathbf{S}_\epsilon^{-1}\mathbf{K}$ and $\mathbf{S}_a^{-1}$ respectively. For the state vector component with a strong *a priori* constraint, i.e., a small variance in $\mathbf{S}_a$, the retrieved result gravitates towards the value of the *a priori* state





vector, while for the one with a weak constraint, i.e., a high variance in $\mathbf{S}_a$, its retrieved value is primarily determined from the measurements.

The variance of a retrieved parameter is equal to the corresponding diagonal element of the covariance matrix $(\mathbf{S}_a^{-1} + \mathbf{K}^T\mathbf{S}_\epsilon^{-1}\mathbf{K})^{-1}$ (see Eq. 23), thus less or equal to the corresponding *a priori* variance in the *a priori* $\mathbf{S}_a$ matrix. In other words,

the change magnitude of a retrieval parameter at each iteration is usually smaller than its *a priori* standard deviation. Consequently the OE method can be used as an inversion scheme to ensure retrieval stability and preserve the dependence of the retrieved results on the measurements, through a careful construction of the *a priori* covariance matrix $\mathbf{S}_a$. To further reduce the dependence on the *a priori* state vector, it is updated at each iteration with the linearization point, setting $\mathbf{x}_a = \mathbf{x}_L$, and hence Eq. (22) becomes

$$\Delta\mathbf{x} = (\mathbf{S}_a^{-1} + \mathbf{K}^T\mathbf{S}_\epsilon^{-1}\mathbf{K})^{-1}\mathbf{K}^T\mathbf{S}_\epsilon^{-1}\Delta\mathbf{y}_L = \mathbf{G}\Delta\mathbf{y}_L, \qquad (25)$$

where $\mathbf{G} = (\mathbf{S}_a^{-1} + \mathbf{K}^T\mathbf{S}_\epsilon^{-1}\mathbf{K})^{-1}\mathbf{K}^T\mathbf{S}_\epsilon^{-1}$ is the optimal estimation gain matrix. This setting floats the anchor point of the retrieval, allowing the measurements to drive the iteration to its final state, with the *a priori* covariance to limit the deviation from the anchor.

By relaxing the *a priori* constraints through increasing the diagonal terms (i.e., the variances) of $\mathbf{S}_a$ such that $\mathbf{S}_a^{-1} \to \mathbf{0}$, $\mathbf{G}$

becomes $\mathbf{G}_{DF}$ and Eq. (22), as well as Eq. (25), becomes Eq. (20). In other words, the direct fitting inversion is a special case of the OE inversion scheme, which is more appropriately called the regulated direct fitting inversion. Using the knowledge of their variances ($\mathbf{S}_a$) to limit some of their ranges while allowing others to change freely, the DVCF algorithm with regulated inversion scheme is suitable for retrieving multiple parameters from discrete measurements . It is applied to EPIC UV observations for simultaneous $O_3$ and $SO_2$ retrievals.

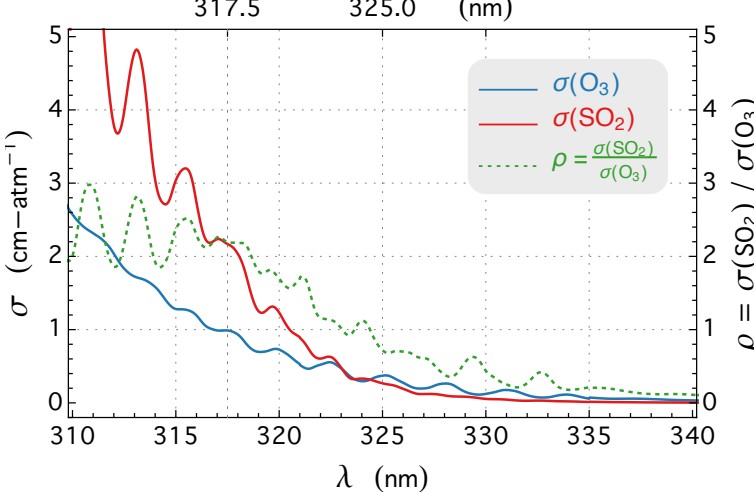

**Figure 13.** EPIC bandpass-averaged cross sections $\sigma$ for $O_3$ and $SO_2$ at 280 Kelvin and their ratio, $\rho=\sigma(SO_2)/\sigma(O_3)$.





## 5 Retrieval from EPIC UV Bands

EPIC have four UV channels (see Fig. 2), referred to as B1, B2, B3, and B4 and characterized by wavelengths $\lambda_1 = 317.5$ nm, $\lambda_2 = 325.0$ nm, $\lambda_3 = 340.0$ nm, and $\lambda_4 = 388.0$ nm, respectively. The radiance measurements from shorter UV channels, EPIC B1 and B2, are sensitive to both $O_3$ and $SO_2$ absorptions (see Fig. 13), containing information that allow the retrieval of

total $O_3$ and $SO_2$ vertical columns, provided that the reflectivity of the underlying surface is known. This knowledge is obtained from the radiance measurements of EPIC B3 and B4, the longer wavelength channels. These channels provide information about the surface reflection and particle back-scattering and have very low sensitivities to $O_3$ and $SO_2$ absorption, such that changes in $O_3$ and $SO_2$ amounts result in little difference in the radiance measurements of these two bands. The reflectivity determined from B3 and B4 is used to estimate the reflectivity at the shorter wavelength ($O_3$ sensitive) channels, accomplished

with the $r_s f_c$ extrapolation scheme (see section 2.3). The reflectivity spectral slope $c_l$ of this extrapolation is proportional to the AI (see Eq. 11). The reflectivity parameter ($R$) is either the LER value $r_s$ estimated from Eq. (6) or the MLER cloud fraction $f_c$ from Eq. (9) depending on the value of $f_c$: $R = r_s$ when $f_c = 0$, $R = f_c$ when $f_c > 0$, and its spectral slope is calculated as $c_l = (R_4 - R_3)/(\lambda_4 - \lambda_3)$.

In this section, we describe the application of the DVCF algorithm to EPIC UV measurements, the scheme to solve the

difficulty arisen from the non-coincidence among the different EPIC spectral observations, and examples to illustrate the success of this application.

### 5.1 Reflectivity Correction by Spatial Optimal Estimation (SOE)

The estimation of $O_3$ column from EPIC radiance measurements requires accurate reflectivity information of the underlying surface, which is extrapolated from the reflectivity determined at the longer wavelength bands (B3 and B4), but the uncer-

tainty of this extrapolation becomes large due to EPIC's asynchronous spectral measurements. Unlike most space-borne UV instruments which provide coincident measurements from different spectral bands, EPIC takes the spectral images sequentially, separating by a time delay of ∼30 seconds between adjacent UV bands. Due to the Earth's self-rotation and spacecraft jitter, different spectral images record slightly different (i.e., rotated) sunlit hemispheres. The geolocation procedure of EPIC (Blank, 2017) aligns different spectral images and further refines the band-to-band registration using the image correla-

tion technique (Yang et al., 2000), which is estimated to provide better than 0.1-pixel (a pixel refers to an IFOV) registration accuracy for EPIC bands. Despite this high registration accuracy, $r_s f_c$ extrapolation (see section 2.3 becomes less accurate for a significant fraction of EPIC IFOVs as substantial reflectivity changes may occur with small shifts in viewing and solar zenith angles since near the direct backscattering direction the particle scattering phase functions have a high angular sensitivity and the shadow areas of structured scenes change non-linearly with viewing-illumination geometry. This difficulty is unlikely to

improve even with better alignment and requires a new approach to correct the extrapolated reflectivity.

The basic idea to obtain a more accurate reflectivity at an $O_3$ sensitive band is to derive it from the radiance measurement of this band with an optimally estimated total $O_3$ column from the nearby $O_3$ distribution. This $O_3$ estimation is attainable because an actual spatial distribution of total $O_3$ column is a smooth function of geolocation and exhibits a high degree of close-range





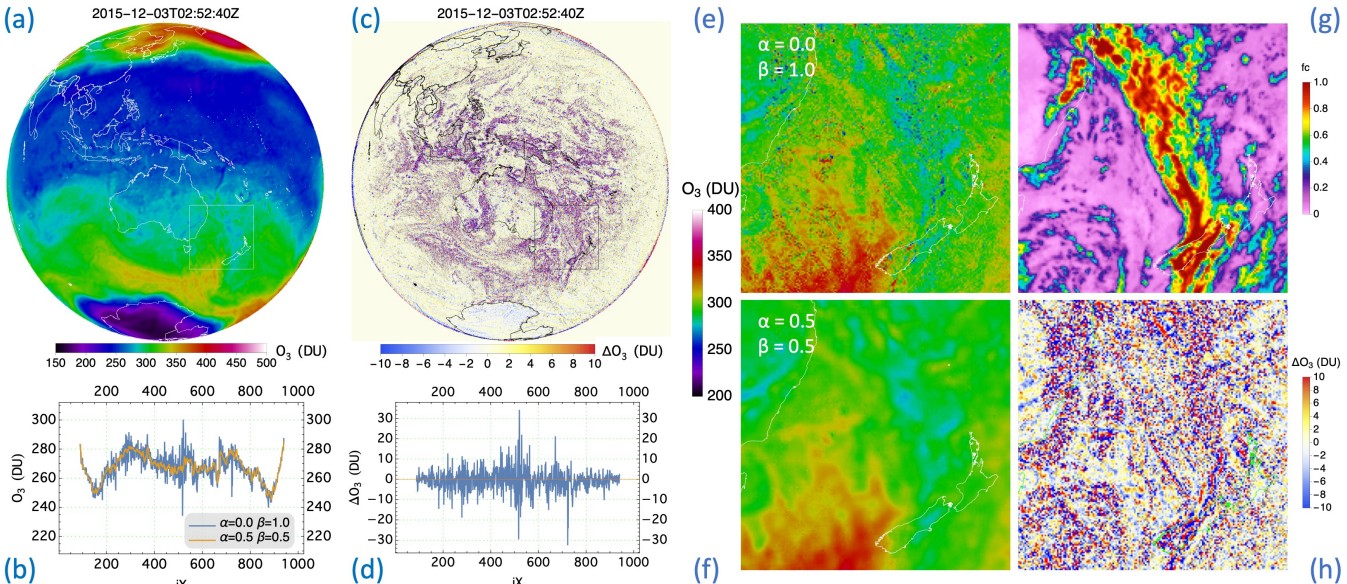

**Figure 14.** Retrieved $O_3$ from EPIC measurements of bands B1, B3, and B4 on December 3, 2015. (a) Optimized (i.e., $\alpha = 0.5, \beta = 0.5$) $O_3$ map based on SOE method; (b) a comparison of optimized (orange) and independent-pixel (blue, $\alpha = 0, \beta = 1$) $O_3$ along the horizontal line (left-to-right) across the middle of the $O_3$ map in (a); (c) the $O_3$ difference map: $\Delta O_3 = O_3(\text{Optimized}) - O_3(\text{IndependentPixel})$; (d) the $O_3$ difference along the horizontal line across the middle of the map in (c); (e) a zoom-in of the independent-pixel $O_3$ map; (f) the optimized $O_3$ corresponding to the rectangle in (a); (g) cloud fraction $f_c$ corresponding to (e) and (f); (h) $O_3$ difference (Optimized – Independent-pixel), a zoom-in corresponding to the rectangle in (c).

correction (Liu et al., 2009). Algebraically, the Spatial Optimal Estimation (SOE) method finds the reflectivity ($R_B$) at EPIC band B by minimizing the cost function that embodies the *a priori* knowledge of $R_B$ and $O_3$ spatial distribution. The first part of cost function supports a smooth (i.e., homogeneous) $O_3$ distribution, while the second part penalizes the difference between $R_B$ and its *a priori* value, which is the extrapolated reflectivity ($R_E$) from the longer wavelength EPIC bands. Hence the cost

5   function is written as

$$\Upsilon = \alpha \sum_{\substack{i=1 \\ j=1}}^{n,n} wt(i,j) \left[ \frac{\Omega(i) - \Omega(j)}{\langle \Omega \rangle} \right]^2 + \beta \sum_{i=1}^{n} \left[ \frac{R_B(i) - R_E(i)}{R_E(i)} \right]^2 \tag{26}$$

$$= \sum_{i=1}^{n} \left( \alpha \sum_{j=1}^{n} wt(i,j) \left[ \frac{\Omega(i) - \Omega(j)}{\langle \Omega \rangle} \right]^2 + \beta \left[ \frac{R_B(i) - R_E(i)}{R_E(i)} \right]^2 \right) = \sum_{i=1}^{n} \Upsilon_i, \tag{27}$$





subject to the measurement constraints $\{I_M(\lambda_B, i) = I_{TOA}(\Omega(i), R_B(i), \lambda_B)$, IOFV index $i = 1 \ldots n$, the size of the IFOV group}, which is linearized to become

$$
\begin{aligned}
\Omega &= \Omega(R_E) + \left( \left. \frac{\partial \ln I_{TOA}}{\partial R_B} \right|_{R_B=R_E} \middle/ \left. \frac{\partial \ln I_{TOA}}{\partial \Omega} \right|_{\Omega=\Omega(R_E)} \right) (R_B - R_E) \\
&= \Omega(R_E) + \left. \frac{\partial \Omega}{\partial R_B} \right|_{R_B=R_E} (R_B - R_E) = \Omega(R_E) + S(R_B - R_E)
\end{aligned}
\tag{28}
$$

where $S = \left. \frac{\partial \Omega}{\partial R_B} \right|_{R_B=R_E}$. The IFOV index $i$ is dropped in Eq. (28) without losing clarity. Here $j$ is also an index, labeling the pairing (or other) IFOV in the group and $wt(i,j)$ is the weighting factor that depends on the distance between the $i, j$ pair. $\langle \Omega \rangle$ is the average $O_3$ column for the group. Given $R_E$, which is the band B reflectivity extrapolated from the longer wavelength bands, the total $O_3$ column $\Omega(R_E)$ is retrieved from band B radiance measurement using the exact solution method (see section 4.1), and the associated $O_3$ profile is the column-dependent M2TCO3 climatological profile $\mathbf{X}_m(\Omega)$. The equation of

measurement constraint (Eq. 28) describes a positive (since $S > 0$ usually) linear relationship between total $O_3$ column $\Omega$ and the surface reflectivity (in the neighborhood of $R_E$), increasing $R_B$ requires more $O_3$ absorption to maintain $I_M = I_{TOA}$.

   Minimizing only the first r.h.s term of Eq. (26) leads to the same $O_3$ column for all the IFOVs (i.e., $\{\Omega(i) = \langle \Omega \rangle, i = 1 \ldots n\}$), while minimizing only the second term makes $R_B = R_E$ for each IFOVs. The constants $\alpha$ and $\beta$ are weights to emphasize respectively the smoothness of $O_3$ spatial distribution and the closeness of reflectivity between extrapolation and estimation.

In the SOE scheme, weights are $\alpha = 0$ and $\beta = 1$ for the traditional $O_3$ retrieval, also referred to as independent-pixel retrieval, while for optimized retrieval, equal weights $\alpha = \beta = 0.5$ are used.

   For optimized retrieval, the minimization of the cost function $\Upsilon$ (Eq. 26) can be accomplished by iteratively finding $R_B(i)$ to minimizing each component $\Upsilon_i$. The solution $R_B(i)$ that minimizes $\Upsilon_i$ is found by solving this equation

$$
\frac{\partial \Upsilon_i}{\partial R_B(i)} = \beta \frac{R_B(i) - R_E(i)}{R_E^2(i)} + \alpha \sum_{j=1}^{n} wt(i,j) \frac{(\Omega(i) - \Omega(j)) S_i}{\langle \Omega \rangle^2} = 0,
\tag{29}
$$

which yields

$$
R_B(i) = R_E(i) - \frac{\alpha \, R_E^2(i) \, S_i \left( n' \Omega(i, R_E(i)) - \sum_{j=1}^{n'} \Omega(j) \right)}{\beta \, \langle \Omega \rangle^2 + \alpha \, n' \, R_E^2(i) \, S_i^2}.
\tag{30}
$$

From Eq. (29) to Eq. (30), only the $n'$ nearby IFOVs are included, i.e., $wt(i,j) = 1$ for $i$-$j$ separation within a few ($< 4$) adjacent IFOVs, otherwise $wt(i,j) = 0$. At the start of iteration, $\{\Omega(j) = \Omega(j, R_E), 1 \ldots n\}$, and they are then updated using Eq. (28) with $R_B(i)$ from Eq. (30) for the next iteration, which stops until changes in $\{R_B(i), i = 1 \ldots n\}$ becomes sufficiently

small. In practice, no more than a couple of iterations are needed to reach convergence.

   Figure 14 shows an example of simultaneous retrieval from the IFOVs of an EPIC hemispheric view using the SOE method. The high variability $O_3$ map (Fig. 14e) from the independent pixel retrieval contains many artifacts (high spikes and low dips in $O_3$ columns), which are substantially reduced using the SOE method, resulting in a much more realistic (smooth) $O_3$ map (Fig. 14f). The $O_3$ differences ($\Delta\Omega$) between optimized and independent-pixel retrievals (see Fig. 14c, d, and h) illustrate the



quantitative improvements, with a small mean $O_3$ difference (mean $\Delta\Omega$ within $\pm0.5$ DU) and a sizeable reduction in $O_3$ noise level (standard deviation of $\Delta\Omega \approx 7$ DU). The corresponding reflectivity corrections are quite significant $\sim 0.02$ on average, with a maximum of $\sim 0.1$ deviation from the $r_s f_c$ extrapolations.

In summary, the SOE method performs single band (B1 or B2) multiple IFOVs (or image-based) retrieval, yielding reflec-
tivity ($R$) and total $O_3$ column ($\Omega$), with the associated profile determined by the $O_3$ model (Eq. 12) that retains only the column-dependent M2TCO3 climatological profile $\mathbf{X}_m$. The *a priori* knowledge of the $O_3$ distribution, which is spatially smooth, provides the extra information to correct the initial reflectivity estimation extrapolated from the characterization based on the longer wavelength bands.

## 5.2   Total $O_3$ Column

Radiance measurements of EPIC B1 and B2 radiances have $O_3$ profile sensitivity, which is higher at B1 than at B2, especially at high zenith (SZA or VZA or both) angles (as illustrated in Fig. 3). Compared to the measurement of a single $O_3$ sensitive band, both bands jointly provide more information that allows the refinement of climatological representation of the $O_3$ profile. This refinement is performed by adjusting the climatological profile with the most probable modal profile ($\mathbf{e}_1$, see Eq. 12) so that both B1 and B2 yield the same total $O_3$ column.

For retrieval from EPIC, the full state vector to be inverted is $\mathbf{x} = \{\Omega_0, \gamma_1, \Xi, R_1, R_2\}$, where $\Omega_0$ is the total $O_3$ column, $\gamma_1$ the $O_3$ profile adjustment factor, $\Xi$ the total vertical $SO_2$ column, $R_1$, $R_2$ the MLER parameters at EPIC B1 and B2. The regulated direct fitting of EPIC B1 and B2 radiances is applied to obtain retrieved full state vector $\mathbf{x}$ .

For each IFOV of EPIC, the $O_3$ vertical column is estimated first assuming there is no $SO_2$. The iteration starts with an initial state vector $\mathbf{x}_0 = \{\Omega_0 = \Omega_c, \gamma_1 = 0, \Xi = 0, R_1 = R_1^S, R_2 = R_2^S\}$, where $\Omega_c$ is the climatological total column selected
from the M2TCO3 climatology based on time and location. $R_1^S$ and $R_2^S$ are the corrected MLER parameters at B1 and B2 respectively using the SOE method (see section 5.1).

Since EPIC radiance measurement errors between any two bands are not correlated, the measurement error covariance matrix is diagonal: $\mathbf{S}_\epsilon = \mathrm{diag}(\sigma_{B1}^2 = 0.00345^2, \sigma_{B2}^2 = 0.00345^2)$, estimated from the random errors of the radiance ($I_M$) measurements (see section 6.2).

There is no correlation among retrieval parameters: total $O_3$ column ($\Omega_0$), the deviation ($\omega_1$) of $O_3$ profile from the mean, $SO_2$ column ($\Xi$), and the MLER parameters $R$, except between $R_1$ and $R_2$. The diagonal elements of the *a priori* covariance matrix are $\mathbf{S}_a = \mathrm{diag}(\varepsilon_{\Omega_0}^2 = 10^2 \ \mathrm{DU}^2, \varepsilon_{\gamma_1}^2 = 2^2 \ \mathrm{DU}^2, \varepsilon_{\Xi}^2 = 0.0001^2 \ \mathrm{DU}^2, \ \varepsilon_{R1}^2 = 0.001^2, \varepsilon_{R2}^2 = 0.001^2)$. The off-diagonal elements are equal to zero, $\{S_a(i,j) = 0, \text{when } i \neq j\}$ , except for the elements associated with $R_1$ and $R_2$, which may be set at $S_a(4,5) = S_a(5,4) = 0.98 \ \varepsilon_{R1} \ \varepsilon_{R2} = 0.0099^2$, representing a high degree of correlation (0.99) between $R_1$ and $R_2$.
This $\mathbf{S}_a$ essentially limits the adjustments at each iteration: $|\Delta\Omega_0| \lesssim \varepsilon_{\Omega_0}(10 \ \mathrm{DU})$, $|\Delta\omega_1| \lesssim \varepsilon_{\gamma_1}(2 \ \mathrm{DU})$, $|\Delta\Xi| \lesssim \varepsilon_{\Xi}(10^{-4} \ \mathrm{DU})$, $|\Delta R_1| \lesssim \varepsilon_{R1}(0.001)$, and $|\Delta R_2| \lesssim \varepsilon_{R2}(0.001)$. The strong constraint on $SO_2$ ensures that its column $\Xi$ never deviates far ($> 0.01$ DU) from its initial value 0 during the iteration, essentially enforcing an $SO_2$-free retrieval. The strong constraints on $R_1$ and $R_2$ also ensures that they remain nearly the same as their initial values $R_1^S$ and $R_2^S$. The constraints on $O_3$ parameters

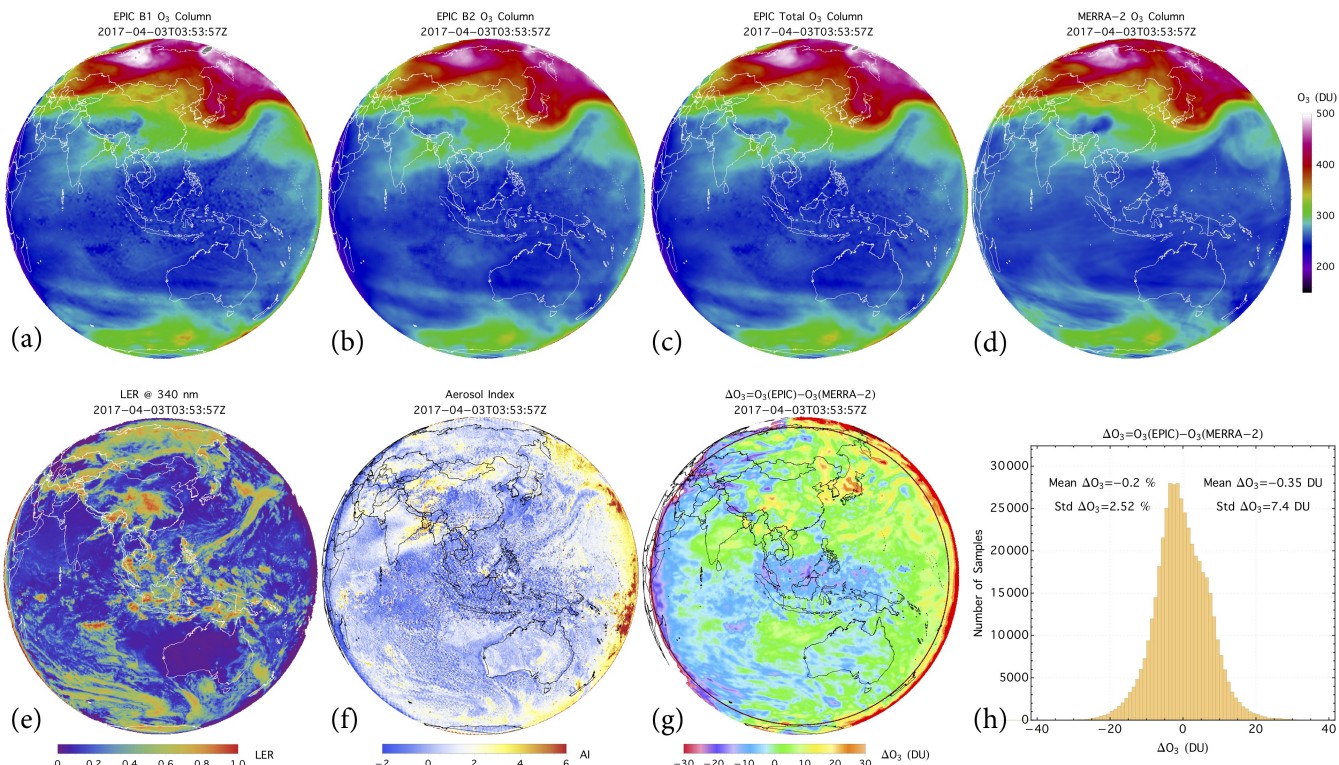

**Figure 15.** An L2 O3SO2AI granule contains the total $O_3$ vertical columns (c), LER at 340 nm (e), and AI (f), retrieved from EPIC UV measurements at 03:53:57 UTC on 04/03/2017. (a) Total $O_3$ column (referred to as B1 total $O_3$ column) retrieved from EPIC B1, B3, and B4. (b) Total $O_3$ column (referred to as B2 total $O_3$ column) retrieved from EPIC B2, B3, and B4. (c) Total $O_3$ from all four bands. (d) Coincident MERRA-2 total $O_3$ columns. (g) The total $O_3$ difference: $O_3(EPIC) - O_3(MERRA-2)$. (h) The histogram of the $O_3$ differences with SZA $\leq 70°$, i.e., samples within the circle in g, with a mean difference $\mu(EPIC) = -0.20\%$ (or $-0.35$ DU) and a standard deviation $\sigma(EPIC) = 2.52\%$ (or 7.4 DU). Similarly the $O_3$ difference, $O_3(EPIC B1) - O_3(MERRA-2)$, has a mean of $\mu(EPIC B1) = 0.25\%$ (or 1.03 DU) and a standard deviation $\sigma(EPIC B1) = 2.68\%$ (or 7.9 DU), and $O_3(EPIC B2) - O_3(MERRA-2)$ has a mean $\mu(EPIC B2) = -0.41\%$ (or $-1.08$ DU) and a standard deviation $\sigma(EPIC B2) = 2.68\%$ (or 7.8 DU).

are quite loose. Especially towards the convergence of the iteration, the absolute adjustment of each component is much smaller than the corresponding standard deviation, i.e., the square root of the corresponding diagonal element of $\mathbf{S}_a$.

With the setup of error and *a priori* covariance matrices $\mathbf{S}_\epsilon$ and $\mathbf{S}_a$, the initial state vector $\mathbf{x}_0$ is updated (Eq. 17) iteratively using $\Delta\mathbf{x}$ from Eq. (25), until the exit of the iteration when $|\Delta\Omega_0| < 0.5$ DU and $|\Delta\gamma_1| < 0.5$ DU. The retrieved total $O_3$ column ($\Omega$) is obtained by integrating the profile $\mathbf{X} = \mathbf{X_m}(\Omega_0) + \gamma_1 \mathbf{e}_1(\Omega_0)$. In processing EPIC data, the initial $O_3$ column $\Omega_c$ of $\mathbf{x}_0$ for an IFOV may be set to the column $\Omega$ of a previous (or nearby) IFOV to improve the speed of convergence of the iteration.





---

**Algorithm 1** : EPIC Total Ozone Retrieval

---

**Data:** EPIC L1B granule containing UV radiances, geolocation, viewing and solar zenith and azimuthal angles

**Climatology:** Minimum surface LER, optical centroid pressure, snow/ice coverage, ozone and temperature profile

**for** Each IFOV **do**

  Determine $R_3$ from B3 radiance and $R_4$ from B4 radiance (Eqs. 6 and 9)

  Compute $c_l$ from $R_3$ and $R_4$, and AI (Eq. 11)

  Extrapolate linearly $R_3$ and $R_4$ to determine $R_{E1}$ at B1 and $R_{E2}$ at B2

  Determine total column $\Omega_{E1}$ from B1 radiance with $R_{E1}$ and $\Omega_{E2}$ from B2 radiance with $R_{E2}$ (Eq. 16)

  Set $R_1 = R_{E1}$, $R_2 = R_{E2}$, $\Omega_1 = \Omega_{E1}$, and $\Omega_2 = \Omega_{E2}$

**end for**

Iteration=0

**while** Iteration < Niter (e.g., Niter = 2) **do**

  **for** Each IFOV **do**

    Update $R_1$ with $R_{E1}$ and $\Omega_1$, and $R_2$ with $R_{E2}$ and $\Omega_2$ (Eq. 30)

  **end for**

  **for** Each IFOV **do**

    Update $\Omega_1$ with $R_1$, $R_{E1}$, and $\Omega_{E1}$, and $\Omega_2$ with $R_2$, $R_{E2}$, and $\Omega_{E2}$ (Eq. 28)

  **end for**

  Iteration = Iteration + 1

**end while**

**for** Each IFOV **do**

  Determine $\Omega$ from B1 and B2 radiances with corrected $R_1$ and $R_2$ (Eq. 25)

**end for**

**Output :** L2 O3SO2AI granule with $\Omega$, AI, and $R_{1...4}$

---

Here we list the algorithmic procedure (see Algorithm 1), titled EPIC Total Ozone Retrieval, that is applied to each EPIC level-1b (L1B) granule, which contains spectral measurements, as well as geolocation and angular information of all the IFOVs of a snapshot of the sunlit side of the Earth, to produce the level-2 (L2) O3SO2AI product. The contents of a sample O3SO2AI granule are displayed in Fig. 15, including total $O_3$, LER, and AI, respectively shown in panels (c), (e), and (f). For

5   comparison, the MERRA-2 assimilated $O_3$ total columns, interpolated to the time and location of EPIC IFOVs, are included in Fig. 15(d), their differences $O_3(\text{EPIC}) - O_3(\text{MERRA-2})$ in (g) and the histogram in (h). This comparison reveals excellent agreement between MERRA-2 and EPIC total $O_3$, showing near identical $O_3$ spatial distributions with similar highs and lows. Quantitatively, the differences for samples with VZA$\leq 70°$ are characterized by a low mean offset ( $\mu(\text{EPIC}) = -0.2\%$ ) and a narrow spread (standard deviation $\sigma(\text{EPIC}) = 2.52\%$). Figure 15 includes the intermediate results of the EPIC total

10   $O_3$ processing (see the procedure in Algorithm 1), showing the total $O_3$ columns retrieved from B1 in panel (a) and from





B2 in panel (b) using the SOE method. Both B1 and B2 total columns closely resemble the MERRA-2 (d) and EPIC (c) total $O_3$ fields, with difference statistics showing slightly worse offsets ($\mu(\text{EPIC B1}) = 0.25\%$ and $\mu(\text{EPIC B2}) = -0.41\%$) and higher standard deviations ($\sigma(\text{EPIC B1}) = 2.68\%$ and $\sigma(\text{EPIC B2}) = 2.68\%$). The improved agreement with MERRA-2 is significant, reducing the B1 $O_3$ spread by $\sqrt{\sigma^2(\text{EPIC B1}) - \sigma^2(\text{EPIC})} = 0.9\%$ (or 2.8 DU) and the B2 $O_3$ spread by

a similar amount. These better agreements are consistent over time and location, substantiating the improved retrieval with both $O_3$-sensitive bands over a single one, which is adopted by the TOMS-V8 algorithm. Since MERRA-2 $O_3$ field, from the assimilation of independent measurements of the Aura OMI and Aura Microwave Limb Sounder (MLS), provides highly realistic spatiotemporal $O_3$ representation, the smaller spread between the two-band (B1 and B2) EPIC and MERRA-2 total $O_3$ columns indicates that the inclusion of more $O_3$ sensitive bands enables more accurate retrievals.

## 5.3   Volcanic $SO_2$

EPIC B1 and B2 radiances respond to both $O_3$ and $SO_2$ absorptions, but with very different (see Fig. 13) sensitivities: $SO_2$ is more than twice as UV-absorbent as $O_3$ at B1, in contrast, it is significantly less at B2, about 70% as absorbent as $O_3$. Consequently, the estimate of $O_3$ absorption signals at these two bands would result in an error due to the presence of $SO_2$ in the atmosphere: 1 DU of $SO_2$ would usually yield more than 2 DU $O_3$ error at B1, but only about 0.7 DU error at B2. This

big difference in absorption sensitivities facilitates the detection of $SO_2$ in the atmosphere. Given a radiance SNR of 285:1, the theoretical minimum detectable level of $SO_2$ enhancement is $\sim 0.5$ DU in the upper troposphere and above. However, it is difficult to distinguish $SO_2$ at this minimum level from other changes, such as $O_3$ profile or surface spectral reflectance, since they can induce similar changes in the measured radiances. This difficulty is increased by the EPIC's asynchronous spectral measurements, which may yield spectral variation similar to the response to adding $SO_2$ in the atmosphere. Consequently, low

levels of $SO_2$ elevation can not be reliably detected in EPIC observations. For significant $SO_2$ elevations, typically those from volcanic eruptions, B1 $O_3$ is much higher than B2 $O_3$ (i.e., $\Omega_1 > \Omega_2$) from the total $O_3$ retrieval (described in section 5.2). Adjusting the $O_3$ profile shape or changing the spectral reflectance of the underlying surface usually cannot eliminate this large $O_3$ discrepancy between the two bands. Therefore a high positive value of $\Delta\Omega$ can be used to flag the presence of $SO_2$. Furthermore, a volcanic plume usually occupies a contiguous area with a limited spatial extent. Thus, $\Delta\Omega$ and $\Omega_1$ enhancements

resulted from volcanic $SO_2$ plume occur over a large group of connected IFOVs instead of isolated or a small group of disconnected IFOVs. Based on these characteristics of volcanic $SO_2$ plumes, we describe next an algorithmic procedure to flag IFOVs with $SO_2$ enhancements.

For reliable $SO_2$ detection, the following procedure is applied to identify the presence of $SO_2$ in an IFOV. First, IFOVs of likely $SO_2$ elevations are flagged through spatial analysis of the differential $O_3$ field (i.e., $\Delta\Omega = \Omega_1 - \Omega_2$, EPIC B1 and B2 $O_3$

difference), accomplished through contour mapping to find closed areas of local $\Delta\Omega$ enhancements, i.e., areas within closed contours with $\Omega_1$ considerably higher ($\geq 7$ DU) than the $\Omega_2$ values (e.g., see Fig.16b). The IFOVs within this $\Delta\Omega$ contour likely have a $SO_2$ elevation around 5 DU or above. After finding the minimum ($\Omega_{\min}$) and maximum ($\Omega_{\max}$) values of $\Omega_1$ in the vicinity outside of this $\Delta\Omega$ contour, contour mapping of $\Omega_1$ is performed to find the longest closed contour line with $\Omega_1$ between $\Omega_{\min}$ and $\Omega_{\max}$ (e.g., see Fig.16a). Within this closed $\Omega_1$ contour, IFOVs with likely $SO_2$ enhancements are flagged





when $\Omega_1 > \Omega_2$. This flagging is then extended to the adjacent areas outside of the two contours to identify IFOVs with possible SO$_2$ contamination. For most volcanic plumes, these two contours overlap each other greatly. Including area within the $\Omega_1$ contour and the adjacent outside regions are designed to capture plumes with lower SO$_2$ elevations.

Once detected, the SO$_2$ quantification follows the DVCF retrieval with the initial state and *a priori* covariance setting described next. For the IFOV identified with a SO$_2$ contamination, the initial O$_3$ values are spatially interpolated from background $\Omega_1$ field, $\gamma_1 = 0$, and initial SO$_2$ column $\Xi_0 = \Omega_1 - \Omega_2$, integrated from a vertical profile specified by a generalized distribution function (GDF, Yang et al. 2010) with a width and a center altitude appropriate for the plume. The corresponding elements of the *a priori* covariance matrix are $\mathbf{S}_a = \mathrm{diag}(\varepsilon_{\Omega_0}^2 = 10^2 \, \mathrm{DU}^2, \varepsilon_{\gamma_1}^2 = 2^2 \, \mathrm{DU}^2, \varepsilon_{\Xi}^2 = \Xi_0^2 \, \mathrm{DU}^2)$, i.e., the variances associated with O$_3$ are the same as those for total O$_3$ retrieval, while the SO$_2$ column variance is equal to the square of the initial SO$_2$ estimate ($\Xi_0$), which is a weak constraint to allow $\Xi$ change freely responding to the measurement . Other retrieval settings are kept the same as in the total O$_3$ retrieval described in the previous section. We list below the complete algorithmic procedure (see Algorithm 2), titled EPIC Total SO$_2$ Retrieval, for SO$_2$ detection and quantification from EPIC UV observations.

---

**Algorithm 2** : EPIC Total SO$_2$ Retrieval

**L1 Data:** EPIC L1B granule containing UV radiances, geolocation, viewing and solar zenith and azimuthal angles

**Climatology:** Minimum surface LER, optical centroid pressure, snow/ice coverage, ozone and temperature profile

**L2 Data:** $\Omega_{E1}$, $\Omega_{E2}$, $R_{E1}$, and $R_{E2}$ from EPIC total O$_3$ retrieval (see Algorithm 1)

**Procedure:** Mapping of $\Omega_{E1} - \Omega_{E2}$ contours to find the closed contour line (enclosing the largest area) with big ($> 7$ DU) B1 and B2 O$_3$ difference

**Procedure:** Mapping of $\Omega_{E1}$ contours to find the largest closed area intersecting with the $\Delta\Omega$ contour line determined in the previous step

**Procedure:** Flagging IFOVs within the $\Delta\Omega$ and the $\Omega_{E1}$ contours with SO$_2$ presence by the condition $\Omega_{E1} > \Omega_{E2}$.

**Procedure:** Extending the SO$_2$ flagging to areas outside but close (within 20 IFOVs, $\sim$400 km) to the two contours identified in the previous steps

**for** Each flagged IFOV **do**

    Initialize O$_3$, SO$_2$, and their variances

    Determine $\Omega$ and $\Xi$ from B1 and B2 radiances with $R_{E1}$ and $R_{E2}$ (Eq. 25)

**end for**

**Output :** update L2 O3SO2AI granule with additional data fields $\Omega$ and $\Xi$

---

The algorithmic procedure (listed in Algorithm 2) applies to regions where EPIC observes volcanic plumes to produce the EPIC volcanic SO$_2$ product. Figure 16 illustrates the detection and quantification of volcanic SO$_2$ from EPIC observations. Spatial analyses (i.e., contour mappings) of the intermediate results (Figs. 16a and b) of the total O$_3$ processing (see Algorithm 1) provide reliable detection of SO$_2$ elevations. The SO$_2$ flagged IFOVs are then processed with the DVCF algorithm to retrieve total vertical O$_3$ and SO$_2$ columns simultaneously, with results showing in Figs. 16 (c) and (d) respectively. Comparison of the two O$_3$ fields in Figs. 16 shows that the initial O$_3$ elevations (Figs. 16a) due to the presence of SO$_2$ are nearly entirely removed

**Figure 16.** EPIC observation of the volcanic plume on 23 June 2019 from the previous day's eruption of Raikoke volcano (represented by $\Delta$ in each panel) in the central Kuril Islands of Russia. (a) B1 $O_3$ column ($\Omega_1$) from EPIC total ozone retrieval and the elevated $O_3$ contour. (b) B1 and B2 $O_3$ column difference ($\Delta\Omega = \Omega_1 - \Omega_2$) and elevated $\Delta\Omega$ contour. (c) Vertical $O_3$ column from EPIC total $SO_2$ retrieval (see Algorithm 2). (d) Vertical $SO_2$ column from EPIC total $SO_2$ retrieval. (e) $SO_2$ vertical column retrieved from a series of eight consecutive EPIC observations of the Raikoke plume, represented by a 1.5 km thick GDF layer centered at an altitude of 13 km above sea level.

in the final $O_3$ field (Figs. 16c), demonstrating that the combo retrieval of $O_3$ and $SO_2$ achieves consistent $O_3$ values inside and outside of the plumes. The achieved internal consistency indirectly validates the $SO_2$ columns. In Fig. 16e, we show maps of





DVCF retrieved SO$_2$ columns from a series of eight consecutive EPIC observations of the Raikoke plume in Fig. 16 (e), with the maximum SO$_2$ value, the total SO$_2$ mass, and the total area covered by elevated SO$_2$ displayed in each snapshot. These results illustrate the high-cadence observing capability and high-quality SO$_2$ measurements of EPIC.

## 6 Error Analysis

We describe in this section how algorithm physics treatments and various sources contribute to the retrieval uncertainties and provide error estimates of the EPIC O$_3$ and SO$_2$ products.

### 6.1 General Expression

The spectral measurements, represented by a column vector $\mathbf{y}$ of length $m$ (the number of wavelength bands), are written explicitly with all the dependent parameters and possible errors and then expanded with respect to the linearization point $(\mathbf{x}_L)$

$$\mathbf{y} = \ln \mathbf{I}_m = \ln \mathbf{I}_{TOA}(\boldsymbol{\omega}, \boldsymbol{\xi}, \mathbf{b}) + \boldsymbol{\epsilon}_m \tag{31}$$

$$= \ln \mathbf{I}_{TOA}(\boldsymbol{\omega}_L, \boldsymbol{\xi}_L, \mathbf{b}_L) + \mathbf{k}_{\boldsymbol{\omega}}(\boldsymbol{\omega} - \boldsymbol{\omega}_L) + \mathbf{k}_{\boldsymbol{\xi}}(\boldsymbol{\xi} - \boldsymbol{\xi}_L) + \mathbf{k}_{\mathbf{b}}(\mathbf{b} - \mathbf{b}_L) + \boldsymbol{\epsilon}_f + \boldsymbol{\epsilon}_m, \tag{32}$$

where the column vectors of length $n_l$ (the number of atmospheric layers), $\boldsymbol{\omega}$ and $\boldsymbol{\xi}$, represents respectively the actual vertical profiles for O$_3$ and SO$_2$, while $\boldsymbol{\omega}_L$ is the climatological O$_3$ profile equal to $\mathbf{X}_{\mathbf{m}}(\Omega_L) + \gamma_1 \mathbf{e}_1(\Omega_L)$ and $\boldsymbol{\xi}_L$ the prescribed SO$_2$ profile specified by a GDF layer with an integrated vertical column equal to $\Xi_L$. The profile weighting functions,

$\mathbf{k}_{\boldsymbol{\omega}} = \frac{-\partial \ln \mathbf{I}_{TOA}}{\partial \boldsymbol{\omega}}|_{\boldsymbol{\omega}=\boldsymbol{\omega}_L}$ and $\mathbf{k}_{\boldsymbol{\xi}} = \frac{-\partial \ln \mathbf{I}_{TOA}}{\partial \boldsymbol{\xi}}|_{\boldsymbol{\xi}=\boldsymbol{\xi}_L}$, are $m \times n_l$ matrices, with each of its rows equal to the product of absorber cross-sections at one spectral band and the layer AMFs (i.e., the mean photon path lengths through the atmospheric layers). Likewise, $\mathbf{b}$ and $\mathbf{b}_L$ are respectively a set of the exact forward model parameters and those used in the linearization, with the corresponding sensitivity matrix $\mathbf{k}_{\mathbf{b}} = \frac{-\partial \ln \mathbf{I}_{TOA}}{\partial \mathbf{b}}|_{\mathbf{b}=\mathbf{b}_L}$. These forward model parameters may include the spectral-dependent MLER parameters, the ground surface and the OCP cloud pressures, the atmospheric temperature profile, the absorption cross-

sections of O$_3$ and SO$_2$, and the parameters that specify the ISRFs of the spectral bands. The column vector $\boldsymbol{\epsilon}_f$ is the forward modeling errors of the spectral bands, such as the approximate radiative transfer through Earth's spherical atmosphere and the incomplete accounting for RRS contributions. The last term $\boldsymbol{\epsilon}_m$ is a column vector representing the spectral radiance errors of the instrument, including random noises and radiometric calibration biases.

Using the definition $\Delta \mathbf{x} = \mathbf{x} - \mathbf{x}_L$ and putting Eq. (32) into residual $\Delta \mathbf{y}_L$, Eq. (25) is re-written as

$$\mathbf{x} - \mathbf{x}_L = \mathbf{G}\left[\mathbf{k}_{\boldsymbol{\omega}}(\boldsymbol{\omega} - \boldsymbol{\omega}_L) + \mathbf{k}_{\boldsymbol{\xi}}(\boldsymbol{\xi} - \boldsymbol{\xi}_L) + \mathbf{k}_{\mathbf{b}}(\mathbf{b} - \mathbf{b}_L) + \boldsymbol{\epsilon}_f + \boldsymbol{\epsilon}_m\right] \tag{33}$$

$$= \mathbf{A}_{\boldsymbol{\omega}}(\boldsymbol{\omega} - \boldsymbol{\omega}_L) + \mathbf{A}_{\boldsymbol{\xi}}(\boldsymbol{\xi} - \boldsymbol{\xi}_L) + \mathbf{G}\mathbf{k}_{\mathbf{b}}(\mathbf{b} - \mathbf{b}_L) + \mathbf{G}\boldsymbol{\epsilon}_f + \mathbf{G}\boldsymbol{\epsilon}_m, \tag{34}$$

where $\mathbf{A}_{\boldsymbol{\omega}} = \mathbf{G}\mathbf{k}_{\boldsymbol{\omega}}$ and $\mathbf{A}_{\boldsymbol{\xi}} = \mathbf{G}\mathbf{k}_{\boldsymbol{\xi}}$ are the averaging kernels (AKs) for O$_3$ and SO$_2$, respectively. Eq. (34) describes how various error sources, from mismatches in absorber profiles to errors in model and measurement, propagate into the final result





($\mathbf{x}$). The rows associated with the $O_3$ and $SO_2$ columns can be extracted from the vector equation (Eq. 34) and written as

$$\Omega - \Omega_T = (\mathbf{A}_\Omega - \mathbf{1})(\boldsymbol{\omega} - \boldsymbol{\omega}_L) + \mathbf{G}_\Omega \boldsymbol{\epsilon}_\Omega, \tag{35}$$

$$\Xi - \Xi_T = (\mathbf{A}_\Xi - \mathbf{1})(\boldsymbol{\xi} - \boldsymbol{\xi}_L) + \mathbf{G}_\Xi \boldsymbol{\epsilon}_\Xi, \tag{36}$$

after subtracting $\Omega_T - \Omega_L$ and $\Xi_T - \Xi_L$ from the row equations respectively. Here $\Omega_T$ and $\Xi_T$ are the true $O_3$ and $SO_2$
5   columns, integrated from the corresponding true $O_3$ ($\boldsymbol{\omega}$) and $SO_2$ ($\boldsymbol{\xi}$) profiles. $\mathbf{A}_\Omega$ and $\mathbf{G}_\Omega$ are the row vectors associated with
the retrieved $O_3$ column $\Omega$ from the corresponding matrices $\mathbf{A}_{\boldsymbol{\omega}}$ and $\mathbf{G}$. Analogously, $\mathbf{A}_\Xi$ and $\mathbf{G}_\Xi$ are the row vectors related
to the retrieved $SO_2$ column $\Xi$ taken from the matrices $\mathbf{A}_{\boldsymbol{\xi}}$ and $\mathbf{G}$ respectively. The constant row vector $\mathbf{1}$ contains the value
for all its elements. Thus its dot product with a vertical profile (a column vector) is equivalent to the summation of all the
individual layer amounts, yielding the total column. The column vector $\boldsymbol{\epsilon}_\Omega$ represents the total error combined from various
sources impacting the total $O_3$ accuracy, including errors in model parameters $\mathbf{k_b}(\mathbf{b} - \mathbf{b}_L)$, forward modeling $\boldsymbol{\epsilon}_f$, spectral
measurements $\boldsymbol{\epsilon}_m$, and the other absorber $\mathbf{k}_{\boldsymbol{\xi}}(\boldsymbol{\xi} - \boldsymbol{\xi}_L)$. Similarly, $\boldsymbol{\epsilon}_\Xi$ represents the combined total error affecting total $SO_2$
accuracy with the other absorber term being replaced by $\mathbf{k}_{\boldsymbol{\omega}}(\boldsymbol{\omega} - \boldsymbol{\omega}_L)$.

Retrieval errors can be characterized using Eqs. (35) and (36), provided errors from various sources are sufficiently small that
forward modeling responds linearly to these deviations. However, substantial retrieval errors usually are resulted from simpli-
fied physics treatments, which constrain the forward model to be radiative transfer in a molecular atmosphere over Lambertian
surfaces. These errors may be called the AMF errors because the simplified physics treatments can not, in general, reproduce
the paths of photons through the observed atmosphere, even though they enable radiance matching between measurement and
modeling. The deviations of mean paths lead to retrieval errors in $O_3$ and $SO_2$ because the interpretation of measured radi-
ance through radiance matching requires accurate modeling of mean photon paths (i.e., the AMFs). The retrieval errors from
the simplified physics treatment can be estimated using closed-loop tests (i.e., realistic forward modelings and then inverse
retrieval with simplified physics treatments). Next, we provide uncertainty estimates of $O_3$ and $SO_2$ retrievals contributed from
various errors sources and simplified physics treatments.

## 6.2 Uncertainty Estimates

### 6.2.1 Measurement Errors

Errors in EPIC spectral measurements contribute to uncertainties in retrieved $O_3$ and $SO_2$ columns ($\Omega$ and $\Xi$). Taken the terms
associated with radiance errors from Eqs. (35) and (36), retrieval errors are written as

$$\Delta\Omega = \mathbf{G}_\Omega(\boldsymbol{\epsilon}_m + \mathbf{k_R}\Delta\mathbf{R}), \tag{37}$$

$$\Delta\Xi = \mathbf{G}_\Xi(\boldsymbol{\epsilon}_m + \mathbf{k_R}\Delta\mathbf{R}). \tag{38}$$

These equations specify how measurement errors ($\boldsymbol{\epsilon}_m$) of the $O_3$ sensitive bands and the MLER parameter errors ($\Delta\mathbf{R}$) due to
the measurement errors in the weak absorption bands propagate into retrieved vertical columns.

Biases in radiance measurements lead to systematic errors in retrieved vertical columns. While the actual radiance biases are
unknown, they are likely less than 1% for EPIC UV bands. For radiance biases within $\pm$1%, the systematic $O_3$ and $SO_2$ column





errors are within $\pm \sim 15$ DU and $\pm \sim 8$ DU, respectively estimated from Eqs. (37) and (38). These retrieval column errors are primarily controlled by the relative differences of spectral errors without significant dependence on the column amounts or surface reflectance. The retrieval biases vary with observing conditions given the same percentage radiance errors due to gain matrices ($\mathbf{G}_\Omega$ and $\mathbf{G}_\Xi$) depend significantly on viewing and illumination angles.

5    In addition to systematic errors, radiance measurement noises add random errors onto the retrieved columns. Retrieval errors due to random radiance noises (specified with normal distributions) are unbiased, with mean values, $\mu(\Delta\Omega)$ and $\mu(\Delta\Xi)$, close to zero and standard deviations, $\sigma(\Delta\Omega)$ and $\sigma(\Delta\Xi)$, proportional to standard deviations of radiance noises. The signal-to-noise ratios for EPIC UV bands are 290:1 (Herman et al., 2018), equivalent to a noise level (standard deviation) of 0.345% (= 1/290). This level is consistent with high-frequency radiance fluctuations (with standard deviations equal to 0.373%, 0.354%, 0.354%, and 0.368% for B1 to B4, respectively) within cloud-free scenes observed by EPIC. With a setting of equal standard deviations for the four UV bands (i.e., $\epsilon_m = \{0.345\%, 0.345\%, 0.345\%, 0.345\%\}$), the estimated column $O_3$ noise level is $\sigma(\Delta\Omega) \simeq 3.2$ DU at low viewing zenith angles, decreases gradually with higher zenith angles, reaches a minimum of $\sim 1.5$ DU at $\sim 75°$, then rebounds quickly with further increases in zenith angles (see Fig. 17). The noise level of $SO_2$ columns, $\sigma(\Delta\Xi)$, exhibits a similar angular dependence as shown in Fig. 17, primarily following the angular variation of the gain matrix $\mathbf{G}_\Xi$. These angular-dependent column noises are insensitive to the column amounts or the surface reflectance.

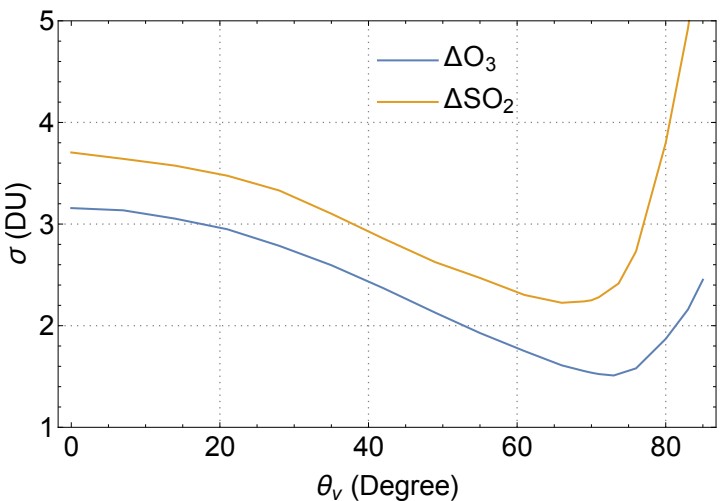

**Figure 17.** Noise levels, i.e., standard deviations ($\sigma$) of $O_3$ and $SO_2$ errors ($\Delta$) contributed from the random noises on EPIC spectral measurements. The $SO_2$ noise estimate is for a layer at an altitude 11 km above sea level.





### 6.2.2 Model Parameter Errors

Retrieval errors due to uncertainties in model parameters, including molecular absorption cross-sections ($\boldsymbol{\sigma}$)and atmospheric temperature profiles ($\mathbf{T}$), are estimated as

$$\Delta\Omega = \mathbf{G}_\Omega \left( (\mathbf{m}_z \Delta \boldsymbol{\sigma}_{O_3})\boldsymbol{\omega} + (\mathbf{m}_z \frac{\partial \boldsymbol{\sigma}_{O_3}}{\partial \mathbf{T}})(\boldsymbol{\omega}\Delta\mathbf{T}) \right), \tag{39}$$

$$\Delta\Xi = \mathbf{G}_\Xi \left( (\mathbf{m}_z \Delta \boldsymbol{\sigma}_{SO_2})\boldsymbol{\xi} + (\mathbf{m}_z \frac{\partial \boldsymbol{\sigma}_{SO_2}}{\partial \mathbf{T}})(\boldsymbol{\xi}\Delta\mathbf{T}) \right), \tag{40}$$

where $(\mathbf{m}_z \Delta \boldsymbol{\sigma}_{O_3,SO_2})$ and $(\mathbf{m}_z \frac{\partial \boldsymbol{\sigma}_{O_3,SO_2}}{\partial \mathbf{T}})$ are $m \times n_l$ matrices with elements $\{m_{z_j}(\lambda_i)\Delta\sigma_{O_3,SO_2}(\lambda_i, T_j), i=1..m, j=1..n_l\}$ and $\{m_{z_j}(\lambda_i)\frac{\partial \sigma_{O_3,SO_2}(\lambda_i,T)}{\partial T}|_{T_j}, i=1..m, j=1..n_l\}$, respectively. Here $m_z$ (see Eq. 7) is the mean photon path length through a layer at altitude $z$, $\Delta\boldsymbol{\sigma}_{O_3,SO_2}$ are errors in $O_3$ or $SO_2$ cross-sections, and $\Delta\mathbf{T}$ are errors in atmospheric temperature profiles.

The BDM $O_3$ cross-sections (Daumont et al., 1992; Brion et al., 1993; Malicet et al., 1995) and the BW $SO_2$ cross-sections (Birk and Wagner, 2018) are used in $O_3$ and $SO_2$ retrievals from EPIC. These baseline cross-sections contain errors, which are not known quantitatively but can be estimated by comparing with alternative cross-sections. Specifically, the BW $O_3$ cross-sections (Birk and Wagner, 2021) and the $SO_2$ absorption cross-sections of Bogumil et al. (2003) are the alternatives that can replace the baselines for EPIC retrievals. The cross-section errors, $\Delta\boldsymbol{\sigma}_{O_3}$ in Eq. (39) and $\Delta\boldsymbol{\sigma}_{SO_2}$ in Eq. (40), are estimated based on the differences between the alternatives and the baselines, showing that alternative $O_3$ and $SO_2$ cross-sections are slightly (about 0.1% to 1.1%) lower than the corresponding baselines at EPIC B1 and B2. These biases in cross-sections result in $O_3$ column biases between 0.5% and 2% and $SO_2$ column biases between 1% and 2% depending on the effective cross-section differences. The temperature-dependence of the BW $O_3$ cross-sections behaves quite differently from the BDM (Bak et al., 2020), especially at EPIC B2, contributing to the high ends of $O_3$ biases ($> 1.0\%$), which occur predominantly at high (viewing or solar) zenith angles when $O_3$ retrieval becomes more sensitive to EPIC B2 radiance.

Both $O_3$ and $SO_2$ cross-sections are significantly dependent on temperatures. Thus accurate temperature profiles are needed to determine atmospheric absorption properties for modeling of measured radiances. As mentioned in section 3, MERRA-2 climatological temperature profiles (Yang and Liu, 2019) are used for retrievals from EPIC. Actual temperature profiles differ from the climatological profiles. Over a short period (e.g., a day), the spatial distribution of these differences is not random, leading to retrieval errors that are unevenly distributed spatially. However, actual temperature profiles are normally distributed around the climatological mean over a long period (e.g., a month) for a location. Therefore temperature profile mismatches add random components, which average to zero over a long time, to the total errors. The variances of these random errors are proportional to the layer-column weighted temperature error variances. Estimated from the variances of temperature profiles (Yang and Liu, 2019), the random components, $\sigma(\Delta\Omega)$, are $\sim$0.3% in the tropics, increase to $\sim$0.7% in the mid-latitudes, and reach $\sim$1% at high latitudes. Similarly, random errors, $\sigma(\Delta\Xi)$, are $\sim$0.8% in the tropics, $\sim$1.7% in the mid-latitudes, and $\sim$3.5% at high latitudes.





### 6.2.3  Forward Modeling Errors

The MLER treatment adopted for the retrieval algorithm allows the use of the vector radiative transfer code, TOMRAD, as the forward model to simulate measured radiances and weighting functions. TOMRAD implements Dave's iterative solution (Dave, 1964) with pseudo-spherical approximation (Caudill et al., 1997) to the problem of the transfer of solar radiations through a

molecular atmosphere over a Lambertian surface. The forward modeling with TOMRAD is accurate for EPIC observations around the center of its hemispheric view, with radiance errors ($\epsilon_f$) of all EPIC UV bands less than $\pm 0.2\%$ for VZA $< 50°$. Note that for EPIC observations, each of its IFOVs has similar VZA and SZA (see Fig. 1) with differences VZA$-$SZA $< \pm 9°$. As EPIC observations move towards the edge, the pseudo-spherical model atmosphere deviates more from Earth's spherical atmosphere in accounting for atmospheric attenuation and multiple scattering, resulting in more significant errors in modeled

radiances, whose maximum errors increase to about $\pm 1\%$ at $75°$ VZA and about $\pm 2\%$ at $85°$ VZA (Caudill et al., 1997). RRS corrections are included in the forward modeling, and they are well within $\pm 1\%$ for EPIC UV bands (e.g., see Fig. 10). Incomplete RRS corrections are expected to add less than $\pm 0.1\%$ to the forward modeling errors.

Unlike the calibration biases being insensitive to observing conditions and having no correlation among different bands, the forward modeling errors vary with absorber amounts and surface reflection and over- or underestimate similarly for all the UV

bands depending on the viewing and illumination geometry. How these radiance errors propagate into the retrieved columns can be estimated using Eqs. (37) and (38), with error source terms replaced by $\epsilon_f$ and $\Delta\mathbf{R}$ due to modeling errors in the long-wavelength bands. With the radiance errors estimated above, these equations yield retrieval errors up to $\pm \sim 0.6$ DU and $\pm \sim 0.3$ DU when VZA $< 50°$, increasing to $\pm \sim 1.5$ DU and $\pm \sim 1$ DU at VZA $= 75°$, and $\pm \sim 5$ DU and $\pm \sim 15$ DU at VZA $= 85°$, respectively for $O_3$ and $SO_2$ vertical column errors. These are systematic errors and vary between high and low

biases spatially depending on observing conditions, especially the viewing and illumination geometry.

### 6.2.4  Profile Errors

As described in section 3, a column-dependent $O_3$ profile, whose tropospheric integration matches the climatological tropospheric column, is used to specify the vertical distribution of a retrieved total $O_3$ column. This retrieved profile ($\boldsymbol{\omega}_L$), which represents likely vertical distribution of the retrieved $O_3$ vertical column, differs invariably from the actual profile ($\boldsymbol{\omega}$). The $O_3$

error ($\Delta\Omega$) due to a profile errors ($\boldsymbol{\omega} - \boldsymbol{\omega}_L$) can be quantified using the first term on the r.h.s of Eq. (35), which is regulated by the retrieval AK ($\mathbf{A}_\Omega$). Examples of AKs for EPIC total $O_3$ retrievals are shown in Fig. 18a, illustrating how $\mathbf{A}_\Omega$ changes with viewing geometry. For low VZAs ($< 55°$), $O_3$ AKs are close to 1 above the upper troposphere, and therefore, profile mismatches in this altitude region result in insignificant retrieval errors. However, profile mismatches produce sizeable retrieval errors for high VZAs. In the troposphere, $O_3$ AKs change with surface reflectance in addition to angular dependence. They are

enhanced drastically above highly reflective surfaces (e.g., snow, ice, or bright clouds), indicating that retrieval error becomes more sensitive to $O_3$ profile mismatches above these surfaces. Under cloud-free conditions, $O_3$ AKs drop quickly towards low-reflectivity surfaces, more so at high zenith angles (e.g., see Fig. 18). In short, $O_3$ errors due to profile mismatches primarily

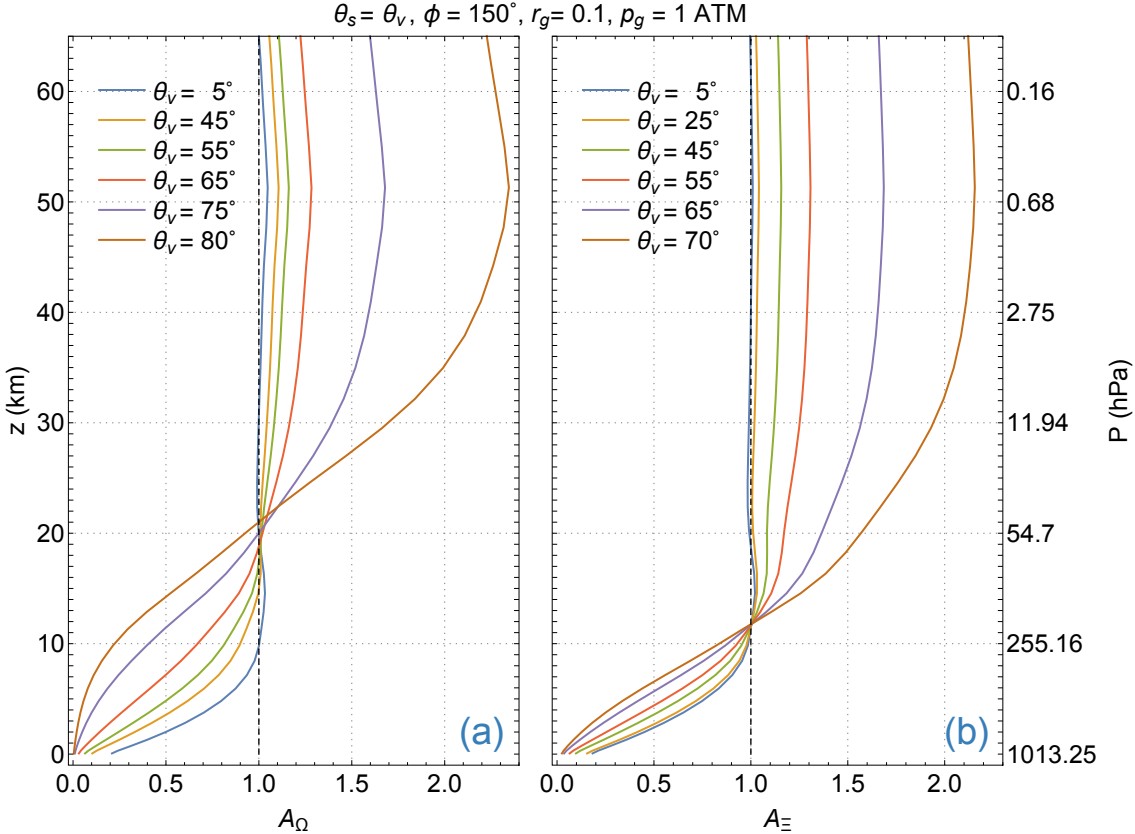

**Figure 18.** Examples of EPIC total $O_3$ AKs (a) and $SO_2$ AKs (b) as functions of geometric altitude ($z$) above seal level for several VZAs ($\theta_v$). These AKs are calculated for a molecular atmosphere at mid-latitude with 275 DU total $O_3$ and 30 DU of $SO_2$ in a layer ( 1.5 km thick) at 11 km altitude over a Lambertian surface. Observing conditions are listed at the top of the figure.

come from the troposphere for low zenith angles, though stratospheric contributions increase substantially with higher zenith angles.

Over a short period (e.g., one day), $O_3$ errors due to profile mismatches are local biases (reductions or enhancements) that vary with location smoothly. However, they are random errors since mismatches ($\omega - \omega_L$) are normally distributed around

5  their near-zero means over a long period (e.g., one month). The variances of $O_3$ errors ($\Delta\Omega$) can be written as

$$\mathrm{Var}(\Delta\Omega) = \mathbf{E}\left[\left((\mathbf{A}_\Omega - \mathbf{1})(\boldsymbol{\omega} - \boldsymbol{\omega}_L)\right)^2\right] = (\mathbf{A}_\Omega - \mathbf{1})\mathbf{E}\left[(\boldsymbol{\omega} - \boldsymbol{\omega}_L)(\boldsymbol{\omega} - \boldsymbol{\omega}_L)^T\right](\mathbf{A}_\Omega - \mathbf{1})^T = (\mathbf{A}_\Omega - \mathbf{1})\mathbf{S}_{n_l}(\mathbf{A}_\Omega - \mathbf{1})^T, \quad (41)$$

where the expected values (i.e., the statistical means), $\mathbf{E}\left[(\boldsymbol{\omega} - \boldsymbol{\omega}_L)(\boldsymbol{\omega} - \boldsymbol{\omega}_L)^T\right]$, are $O_3$ profile covariance matrices $\mathbf{S}_{n_l}$, which depend on total columns ($\Omega$), season, and latitude. This random component is estimated as a function of VZA using the column-dependent $\mathbf{S}_{n_l}$ from the M2TCO3 climatology (Yang and Liu, 2019). Figure 19 shows that the standard deviation of this error

10  component increases gradually with higher VZA, from 1% at nadir to 1.7% at 75°, then rapidly with further elevated VZA.





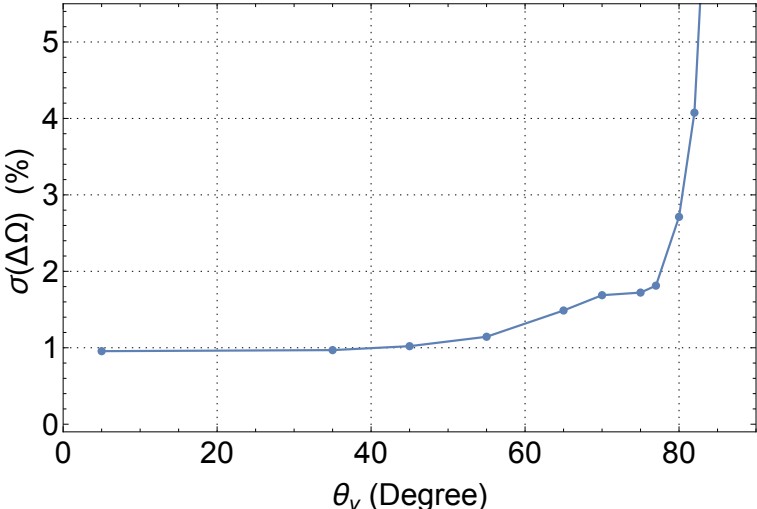

**Figure 19.** The standard deviation ($\sigma$) of $O_3$ errors due to profile mismatches as a function of the viewing zenith angle ($\theta_v$), estimated using the M2TCO3 profile covariance matrix for the December mid-latitude zone (40°N–50°N).

Retrieval of $SO_2$ requires knowledge of the altitude at the center of the volcanic plume, which can be represented by a narrow (e.g., a width of 1.5 km ) GDF. Error in the plume altitude leads to $SO_2$ retrieval error, which can be estimated using the retrieval AK. Figure 18b shows sample AKs and their variations with VZA for an $SO_2$ plume center at 11 km altitude. The values of these AKs are equal to 1 at 11 km, meaning no retrieval error when the altitude used in the retrieval is equal

to the actual plume altitude. When the plume altitude is higher (lower) than the assumed altitude (11 km), the retrieved $SO_2$ column overestimates (underestimates) the actual column. At low VZAs ($< 35°$), AK values are close to 1 above the assumed altitude (11 km), indicating small (less than a few percents) errors for plumes at higher altitudes. Overestimation (upto 150%) increases quickly with larger VZAs when the plume is at higher altitudes. Underestimation is more severe with lower altitude of the plume, by 10 to 20% per 1 km lower than the assumed altitude.

**6.2.5    Errors from Lambertian Treatment of Natural Surfaces**

Reflections from surfaces are anisotropic but treated as isotropic. To estimate errors in $O_3$ and $SO_2$ columns due to this simplification, we performed DVCF retrieval from simulated radiances. First, TOA radiances of the four EPIC UV bands are modeled using a state-of-art radiative transfer model, VLIDORT (Spurr, 2006), for a molecular atmosphere with various $O_3$ and $SO_2$ profiles over a surface characterized by an anisotropic BRDF. Next, GLERs are determined at the long-wavelength

bands (B3 and B4) and then linearly extrapolated to the short-wavelength bands (B1 and B2). Finally, retrieved $O_3$ and $SO_2$ columns from simulated B1 and B2 radiances using the extrapolated GLERs are compared with the column settings of the forward modeling to quantify retrieval errors. Examples of $O_3$ and $SO_2$ errors determined this way are shown in Fig. 20 for observing conditions described in Fig. 4 caption. In the closed-loop testing, surface reflection is specified by the Cox-Munk




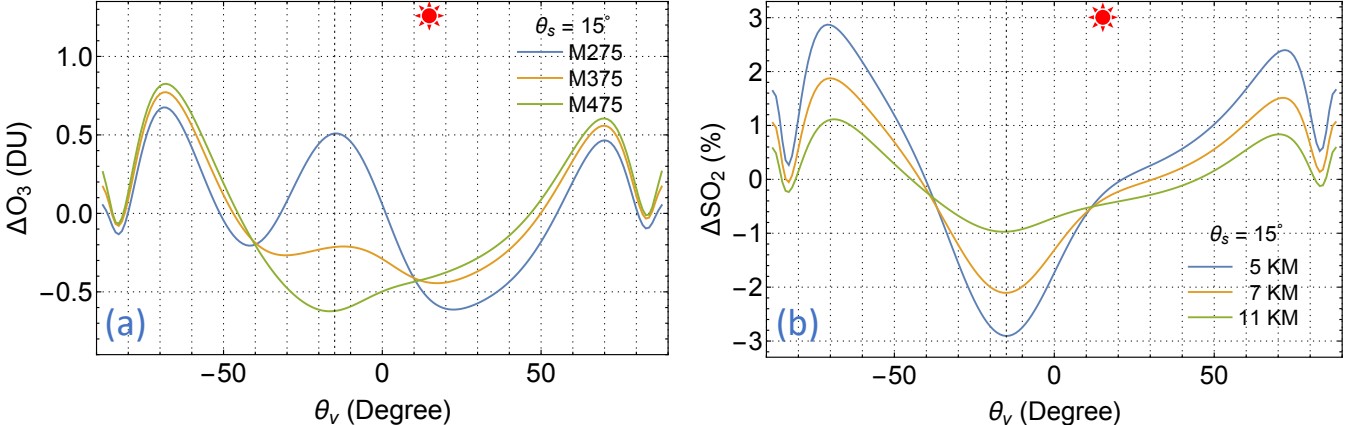

**Figure 20.** Errors in retrieved $O_3$ and $SO_2$ due to Lambertian surface treatment of an anisotropic surface. (a) $O_3$ errors in DU for midlatitude $O_3$ profile with total columns of 275 DU, 375 DU, and 475 DU. (b) $SO_2$ errors in percent for $SO_2$ layer at altitudes of 5 km, 7 km, and 11 km above sea level. See Fig. 4 caption for the specification of surface BRDF and viewing and illumination geometry.

BRDF, which is highly anisotropic, more so than the land surface BRDFs that are well characterized by the combinations of Lambertian, Ross, and Li kernels (Lucht et al., 2000; Schaaf et al., 2011). Hence, the Cox-Munk BRDF selection provides error ranges due to the Lambertian treatment of surface reflections. Closed-loop tests are performed for a wide range of viewing-illumination geometries and vertical distributions of $O_3$ and $SO_2$. Test results (e.g., see Fig. 20) show that errors in total $O_3$ are mostly within $\pm 1$ DU, while $SO_2$ errors are within $\pm 5\%$ percent for $SO_2$ layers above 5 km, and decrease (increase) with higher (lower) layer altitudes. As shown in Fig. 5, the AMF errors due to Lambertian treatment occur below 20 km altitude. Consequently, a small fraction of the $O_3$ profile is affected by this approximation. Thus, $O_3$ errors are proportional to the tropospheric columns but are insensitive to the total column amounts. Since a vast majority of volcanic $SO_2$ clouds are located below 20 km altitude, $SO_2$ errors are proportional to the total $SO_2$ columns. Higher $SO_2$ clouds are not affected by this treatment.

### 6.2.6 Errors from MLER treatment of Clouds and Aerosols

The MLER model is adopted to treat atmospheric particles, including clouds and aerosols, which reside predominantly in the lower troposphere. The modeled light paths (especially in the troposphere) based on this treatment differ significantly from those for light transfer through the particle-laden atmosphere (e.g., see Fig. 9). The retrieval errors due to this simplification are again estimated using closed-loop testing. First, TOA radiances of the EPIC UV bands are simulated using VLIDORT for particle-laden atmospheres with various $O_3$ and $SO_2$ profiles over Lambertian surfaces of different reflectivities. Then inversion from the simulated radiances with the MLER treatment permits the identification of conditions under which retrieval errors are significant.





**Clouds**

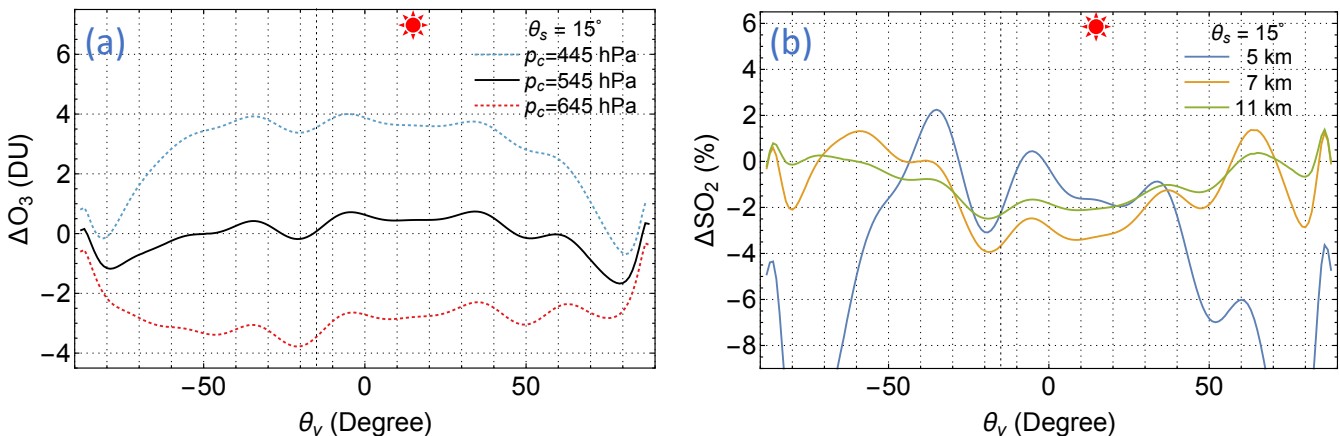

**Figure 21.** Errors in retrieved $O_3$ and $SO_2$ due to MLER treatment of clouds, which are represented by 1.5 km thick C1 particle layers with an optical thickness $\tau = 15$ at 340 nm. (a) $O_3$ errors in DU for correct ($p_c = 545$ hPa) and biased ($p_c = 545 \pm 100$ hPa) cloud OCPs. (b) $SO_2$ errors in percent for $SO_2$ layers at three altitudes (5, 7, and 11 km) above a layer of cloud (at 3 km altitude). See Fig. 4 caption for the specification of viewing and illumination geometry.

The error in total $O_3$ due to the MLER treatment of a low-lying (below 10 km) cloud is mostly within $\pm 2$ DU (e.g., Fig. 21a). This $O_3$ error decreases slightly with a lower cloud altitude (or higher cloud pressure) but is insensitive to the cloud fraction or the total $O_3$ column. In other words, the MLER treatment does not contribute to large uncertainty in the retrieved total $O_3$

column, provided that an accurate OCP for the cloud is used for the MLER cloud surface. However, OCP has some uncertainty, contributing to additional uncertainty in the $O_3$ column: a low (high) bias in OCP results in a positive (negative) error in total $O_3$, quantitatively $\pm 100$ hPa causes about $\mp 4$ DU (see Fig. 21a). The OCP uncertainty is estimated to be within $\pm 50$ hPa, thus contributing $\mp 2$ DU to the total $O_3$ uncertainty. Combining $O_3$ uncertainties due to the OCP error and the MLER treatment yields $\pm 4$ DU uncertainty in total $O_3$ under cloudy conditions.

The error in the total $SO_2$ column due to the MLER cloud treatment is within $\pm 2\%$ when the $SO_2$ layer in the troposphere is well above the underlying cloud. This $SO_2$ error increases with a smaller separation between the $SO_2$ layer and the cloud, reaching $\pm 15\%$ when the $SO_2$ layer is just above the cloud. These characteristics of $SO_2$ error are illustrated in Fig. 21b. In contrast to the MLER-treatment $O_3$ error, which is insensitive to the total column, this $SO_2$ error is proportional to the total $SO_2$ column. When the $SO_2$ layer is below or within the cloud, the uncertainty of $SO_2$ quantification increases drastically.

Depending on the relative distributions of $SO_2$ and the cloud particles, the retrieved $SO_2$ based on the MLER treatment can be a fraction of or a few times the actual column.

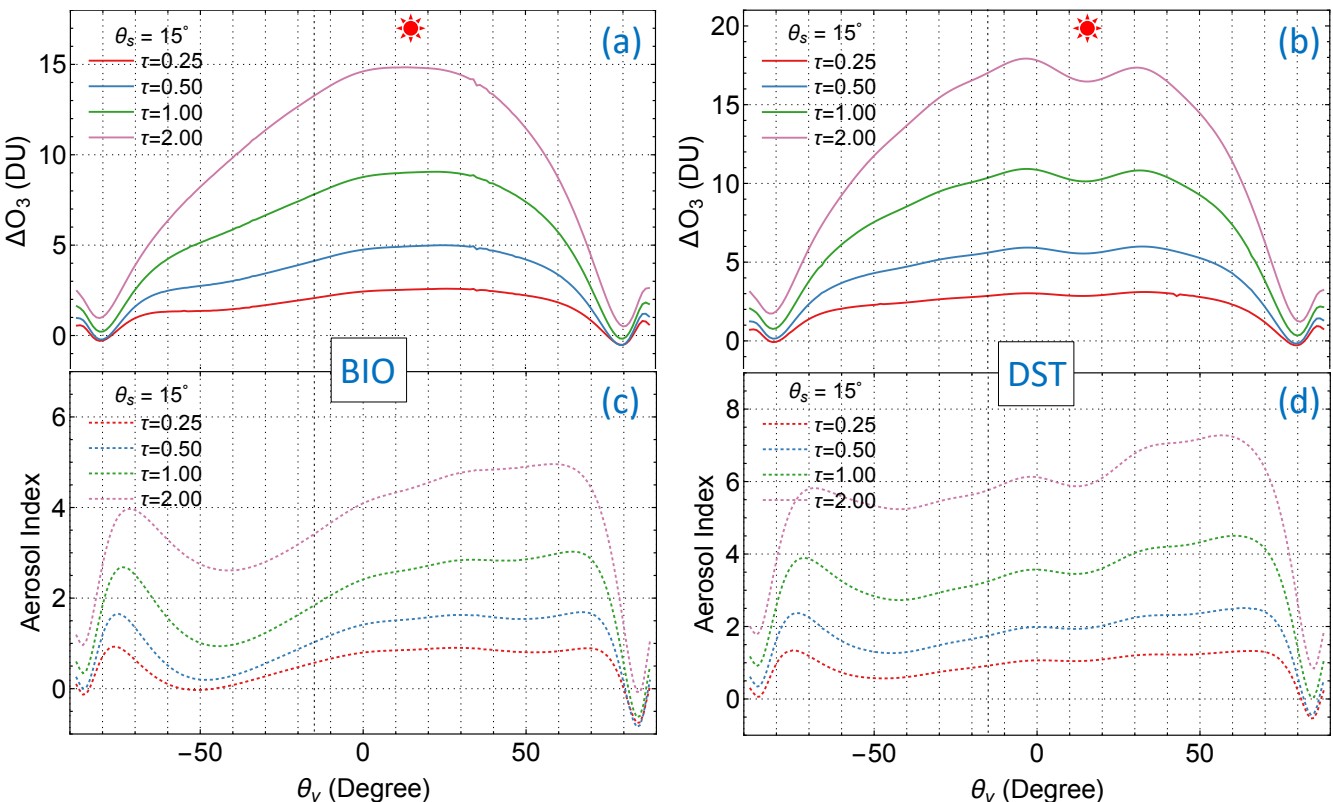

**Figure 22.** Errors in retrieved $O_3$ due to MLER treatment of two common UV-absorbing aerosols, (a) BIO ($\omega = 0.921$) and (b) DST ($\omega = 0.900$), with various optical thicknesses ($\tau = 0.25$, $0.5$, $1.0$, and $2.0$ at 340 nm) located at 5 km altitude. See Fig. 4 caption for the specification of viewing and illumination geometry. The AIs associated with each observation scenario are shown in panels (c) and (d).

**Aerosols**

Besides clouds, the MLER treatment is applied to IFOVs contaminated by aerosols, which reside primarily in the troposphere and cover a significant portion of Earth's surface. These aerosols are suspended tiny (micron-scale) particles that scatter and possibly absorb sunlight. The frequently observed non-absorbing (or weakly absorbing) aerosols are sea salt and sulfate (SLF), and UV-absorbing aerosols are smoke (i.e., carbonaceous aerosols from biomass combustion, BIO), mineral dust (DST), and volcanic ash. Moderate and high positive AI values indicate the presence of UV-absorbing aerosols in an IFOV, while negative and slightly positive AI values the presence of non-absorbing or weakly absorbing aerosols.

Closed-loop testing shows that MLER treatment of non-absorbing and weakly absorbing aerosols reside in the lower troposphere ($< 7$ km) result in small ($< \pm 2$ DU) errors in total $O_3$ retrievals, provided that the proper OCP for the elevated cloud surface is used. This error range is nearly independent of the total $O_3$ column or the aerosol loading.

The MLER treatment errors for UV-absorbing aerosols close to the surface ($< 1$ km altitude) are mostly within $\pm 1$ DU, similar to the error range associated with the LER treatment of BRDF surfaces. For elevated UV-absorbing aerosols, the





MLER treatment and the linear $r_g f_c$ extrapolation scheme (see section 2.3) results in a positive bias in the retrieved total $O_3$ columns (e.g., see Fig. 22). This $O_3$ bias depends on the viewing-illumination geometry and generally increases with stronger aerosol absorption (i.e., lower single scattering albedo, $\omega$), larger aerosol optical thickness, and higher altitude of aerosol layers. Regression analysis of results from closed-testing with many combinations of viewing-illumination geometries, particle-laden

atmospheres (with various optical properties, optical thicknesses, and vertical distributions), surface reflectivities, and $O_3$ profiles reveals a positive correlation between column $O_3$ error and the UV AI. Quantitatively this relationship can be written as $\Delta O_3 = (1.5 \pm 1) \times$ AI DU for AI values greater than 0.5 and less than 8. This relationship provides a rough estimate of $O_3$ bias based on the observed AI. Typically, AI values fall between 1 and 4 with a median value of 1.5 for EPIC observations of UV-absorbing aerosols, corresponding to a mean $O_3$ bias of about 3 DU for IFOVs contaminated with UV-absorbing aerosols.

The MLER treatment sometimes fails when aerosol absorption is strong such that the derived LER becomes negative. In this case, the explicit aerosol treatment may be needed to reduce the retrieval uncertainty.

EPIC's high-cadence observation has more chances to view volcanic clouds during or soon after eruptions. These young volcanic clouds contain mixtures of ash particles and water or ice clouds, as eruptions inject ash and gases (including $SO_2$) into the atmosphere. Since ash particles absorb UV strongly, the MLER treatment of volcanic plumes leads to huge uncertainties in

the retrieved $SO_2$ columns, which are often over- or under-estimated greatly depending on the relative distributions between $SO_2$ and ash particles. An explicit treatment of volcanic ash is needed for accurate retrieval of $SO_2$ when ash particles are co-located with or slightly separated from the gas.

## 7   Validation of EPIC $O_3$ and Comparison of $SO_2$

### 7.1   $O_3$ Validation

We validate the DVCF $O_3$ retrievals from EPIC using ground-based Brewer spectrophotometer measurements and and the assimilated $O_3$ product from MERRA-2, the Modern-Era Retrospective Analysis for Research and Applications, Version 2 (Gelaro et al., 2017).

We compare EPIC total $O_3$ columns with the Brewer $O_3$ data at ten selected ground stations with high-cadence measurements, distributed in five latitude zones. At each of these selected stations, a Brewer spectrometer makes a measurement every

few ($\sim$ 10) minutes during the daylight hours each day, thus providing total vertical $O_3$ columns that are coincident (within $\pm 15$ minutes) with EPIC observations at these stations. For inter-comparison, Brewer $O_3$ data are interpolated to the times when EPIC observes these locations. Coincident $O_3$ data from these two independent sources are displayed in the upper panels of Fig. 23(a-e), their differences in the lower panels of Fig. 23(a-e), and the EPIC vs. Brewer scatter plots in the right panels of Fig. 23(a-e). We include coincident data with VZA $\leq 70°$ only for statistical analysis, due to EPIC-Brewer IFOV differ-

ences that usually increase with slant path lengths and EPIC's footprint sizes and due to calibration biases at large VZAs or SZAs. The mean difference and standard deviations in percent are displayed in the difference plots, while those in DU and the correlation coefficients are in the scatter plots. Time series of $O_3$ difference (see lower panels of Fig. 23(a-e) between EPIC and Brewer are highly stable with similar moving averages and standard deviations from June 2015 to April 2021, showing





**Figure 23.** Inter-comparison of total $O_3$ from EPIC and the ground-based Brewer spectrophotometers at ten selected ground stations with high-cadence measurements: Alert (82.50°N), Eureka (79.99°N), ), Resolute (74.72°N), Churchill (58.75°N), Edmonton (53.55°N), Goose Bay (53.31°N), De Bilt (52.10°N), Thessaloniki (40.63°N), Paramaribo (5.806°N), and South Pole (-89.99°N), from July 2015 – April 2021. EPIC and Brewer coincident pairs are used in the plots and data with VZA $\leq$ 70°N only are included in the difference statistics.

that EPIC $O_3$ are consistent over time, without noticeable drift. The correlations between EPIC and Brewer are very high with the correlation coefficients R $\geq$ 0.96 for most stations (except for the Paramaribo station near the equator, where R = 0.87),



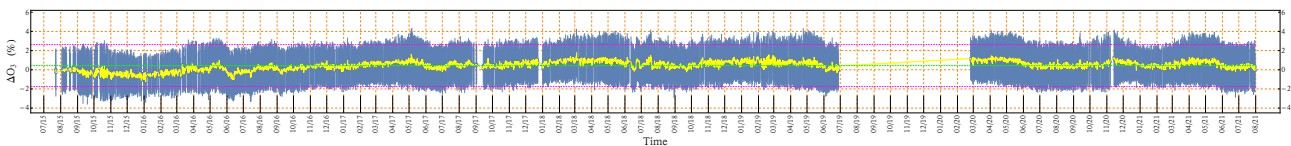

**Figure 24.** Comparison of synoptic EPIC $O_3$ with MERRA-2 assimilated $O_3$: time series of mean daily differences and standard deviations for EPIC observations with VZA $\leq 70°$.

demonstrating that EPIC captures $O_3$ variability accurately. EPIC $O_3$ agrees with the Brewer measurements to better than 1% with standard deviations of differences less than 3.5% for all the ground stations, validating the high accuracy of EPIC total $O_3$.

From October 2004, MERRA-2 $O_3$ field is assimilated from Aura MLS and OMI and provides highly realistic global distri-
butions of $O_3$ in the stratosphere and upper troposphere while inheriting the uncertainty characteristics of its sources (Stajner et al., 2008; Wargan et al., 2015; Davis et al., 2017). We compare the MERRA-2 synoptic $O_3$ field with the EPIC hemispheric view for the same observation time to access EPIC's capability in capturing the realistic $O_3$ distribution. For instance, strikingly similar $O_3$ spatial distributions are observed in EPIC measurements (Fig. 15c) and the MERRA-2 assimilation (Fig. 15d), with an agreement at -0.20 $\pm$ 2.52% (or -0.35 $\pm$ 5.6 DU, Fig. 15h). We expand this synoptic comparison to each EPIC hemispheric
view obtained from July 2015 to August 2021 and plot in Fig. 24 the time series of daily statistics. This time series shows that nearly the same level of agreement is achieved for the entire period, with a mean bias and standard deviation of 0.44 $\pm$ 2.19% (or 1.16 $\pm$ 6.34 DU). Considering the mean bias (about $-1.2\%$) of MERRA-2 total $O_3$ (Wargan et al., 2017), we estimate the accuracy of EPIC total $O_3$ to be $-0.76 \pm 2.19\%$.

## 7.2   SO$_2$ Comparison

Volcanic eruptions occur sporadically and without warning but EPIC on DSCOVR, from the unique L1 vantage point, usually provides multiple daily observations of volcanic $SO_2$ and ash clouds once injected into the atmosphere. In contrast, ground-based instruments rarely detect volcanic clouds unless they drift over one in operation. We thus rely on polar-orbiting instruments, which may observe a volcanic cloud once (or more at high latitude) per day to provide validation measurements.

The OMPS-NM on SNPP provides high-quality hyperspectral measurements in the UV, from which highly accurate retrievals of $O_3$ and $SO_2$ are achieved using the DVCF algorithm. The DVCF algorithm can apply to both discrete spectral measurements (e.g., TOMS and EPIC) as well as hyper-spectral ones (e.g., OMI and OMPS-NM). The main difference is more information can be extracted from hyperspectral measurements to improve the accuracy and precision of the retrieved geophysical parameters. For instance, the altitude of an $SO_2$ layer can be determined in addition to its amount simultaneously using the
DVCF algorithm (Yang et al., 2010). Having the altitude information significantly improves the accuracy of $SO_2$ quantification because the $SO_2$ measurement sensitivity varies strongly with its altitude. Thus DVCF height-resolved $SO_2$ retrievals from hyperspectral instruments, such as OMI and OMPS-NM, provides the most accurate quantification of $SO_2$ vertical columns.

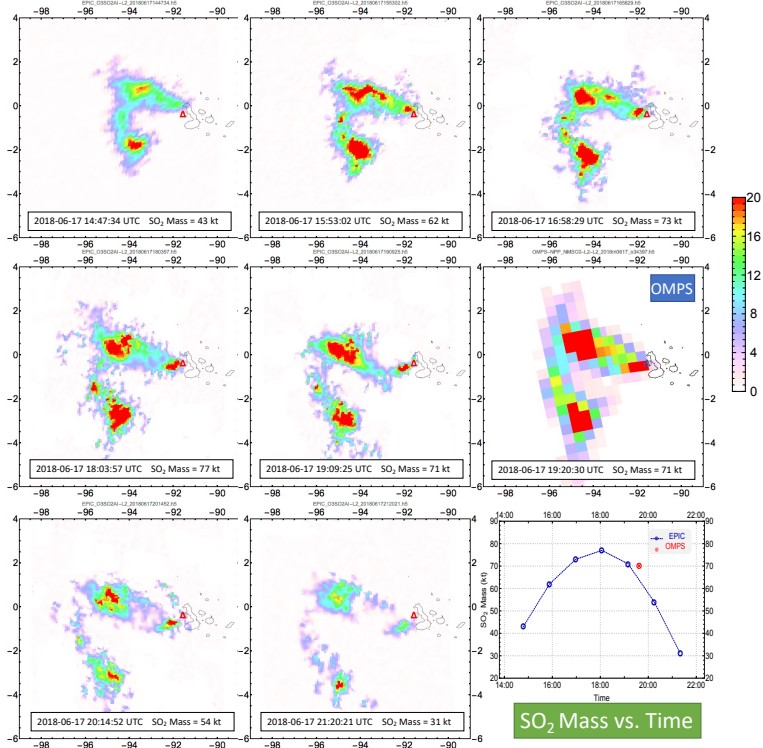

**Figure 25.** EPIC and OMPS observations of volcanic SO$_2$ plumes on 17 June 2018 from the eruption of Fernandina volcano ($\Delta$) in the Galapagos Islands. This eruption injected significant amount on SO$_2$ into the troposphere at about 3.5 km above sea level. The mass loading of a SO$_2$ plume is obtained by summing the SO$_2$ masses of all IFOVs with SO$_2$ vertical columns $\geq$ 1 DU. The lower right panel plots the EPIC and OMPS SO$_2$ masses vs. the observation time (UTC).

To validate DVCF SO$_2$ retrievals from EPIC, we compare the SO$_2$ mass loading of a volcanic plume integrated from EPIC observations with SNPP OMPS-NM for the same event.

Figure 25 compares the DVCF retrievals of the volcanic plume from the explosive eruption of Fernandina volcano in the Galapagos Islands on 17 June 2018. Seven plume exposures about 65 minutes apart are taken by EPIC on this day. Soon
5  after the fifth EPIC exposure, the OMPS observed this plume for the first time. For the exposures at ten minutes apart, both instruments estimate the mass loading at 71 kt, validating the EPIC SO$_2$ result.

The lower right panel of Fig. 25 plots the plume mass vs. the observation time, showing the mass loading peaks near the local noontime. The observed mass change results from the continuing emission from the volcano, the conversion of SO$_2$ into sulfate, and the changing measurement sensitivity with viewing illumination conditions since low SO$_2$ columns may be missed
10  at large (VZA or SZA or both) angles due to lower sensitivity. EPIC's high-cadence observations allow better identification of the peak loading of volcanic SO$_2$ plume, thus usually can provide more accurate estimates of the lower bound of SO$_2$ emission compared to polar-orbiting instruments.



We have conducted many mass loading comparisons between EPIC and OMPS and found that the agreements are usually within 20%. These findings indicate that DVCF $SO_2$ retrieval from EPIC provides better than 20% (an estimate of the upper error bound) accuracy in total mass for eruptions with greater than 50 kt emissions.

## 8   Conclusions

We present the algorithm for making the EPIC O3SO2AI product in this algorithm theoretical basis document (ATBD). This algorithm is based on the DVCF algorithm developed for retrieving trace gases, including $O_3$, $SO_2$, and $NO_2$, from Aura OMI and Suomi NPP and NOAA-20 OMPS. Algorithm advances, including the improved $O_3$ profile representation and the regulated direct fitting inversion technique, improve the accuracy of $O_3$ and $SO_2$ from the multi-channel measurements of DSCOVR EPIC. The theoretical basis of the SOE approach, introduced to reduce retrieval artifacts due to EPIC's band-to-

band misregistration, can be exploited for other applications, such as the separation of a spatially smooth data field (e.g., stratospheric $O_3$) from that (e.g., tropospheric $O_3$) with higher spatial variations.

A thorough error analysis is provided to quantify $O_3$ and $SO_2$ retrieval uncertainties due to various error sources and simplified algorithm physics treatments. Error analysis findings indicate that the MLER treatment of UV-absorbing aerosols leads to significant uncertainties in retrieved $O_3$ and $SO_2$ columns. Future improvements may include explicit aerosol treatment or other

schemes for radiance or product corrections. The GLER treatment of anisotropic surface reflections introduces small errors in the retrieved total $O_3$ and $SO_2$ columns, primarily because surface reflection is a minor component of measured radiance in the UV. However, this GLER treatment does not generally provide a more accurate tropospheric AMF. Hence, explicit BRDF treatment of surface reflection is needed for accurate retrievals of tropospheric gases.

The EPIC total $O_3$ columns are validated against coincident ground-based Brewer measurements and compared with coin-

cident $O_3$ data from MERRA-2 assimilation. The findings show that EPIC total $O_3$ is highly accurate, capturing the short-term $O_3$ variability realistically while maintaining long-term consistency over the entire record. The EPIC $SO_2$ laodings of volcanic plumes are evaluated against those from hyper-spectral measurements of the same eruptions, showing that EPIC provides accurate $SO_2$ quantifications from large volcanic eruptions. EPIC's high-cadence observations allow better identification of the peak loading of volcanic $SO_2$ plume compared to polar-orbiting instruments.

*Data availability.*

The EPIC product, O3SO2AI, contains scene reflectivity, aerosol index (AI), total vertical columns of ozone ($O_3$), and vertical sulfur dioxide ($SO_2$) columns when volcanic clouds are detected in the EPIC field of view. This product available at NASA Langley Atmospheric Science Data Center (ASDC), accessible at this link:

https://asdc.larc.nasa.gov/project/DSCOVR/DSCOVR_EPIC_L2_O3SO2AI_02.



*Author contributions.*

XH (the lead author) contributed to the development of computer codes, conducted data analysis and product validations, and prepared the initial draft of the paper. KY conceived the research, designed the algorithm, contributed to algorithm implementation, data analysis, and product validations, and contributed to writing the article.

5   *Competing interests.*

The authors declare that they have no conflict of interest.

*Acknowledgements.*  This work is supported by NASA.





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
