# Peer review of "Algorithm Theoretical Basis for Ozone and Sulfur Dioxide Retrievals from DSCOVR EPIC"

_Atmospheric Measurement Techniques, 2022_

## Author Comment (AC1)

We like to thank Referee #1 for a positive review and comments. Below, Referee's comments are in blue and our responses in black.

Anonymous Referee #1, 29 Jun 2022
General comments

This paper presents the algorithm details employed for retrieving total columns of O3 and SO2 from DSCOVER EPIC UV measurements with the direct vertical column fitting method. The paper is well structured and written. For general readers, the author kindly describes the radiative transfer process of incoming UV photons and forward model calculations along with the comprehensive sensitivity tests. The inverse process from spectral measurements and geophysical variables are specified with error analysis. And then, retrievals are validated using brewer measurements/MERRA-2 for O3 and OMPS products for SO2. I agree with the conclusion of this paper, the maturity level of the presented algorithm is very high and the EPIC hourly measurements are very promising.

I would like to recommend this article for the publication of ATM after revising serval minor aspects.

Minor comments

1. I think that this article is a good textbook for students who just step into the atmospheric remote sensing area. But, this manuscript needs to be tightened in the format of research article, especially for section 2.

   Response:

   This paper is for a broad range of readers, from novices to experts in the field. The included contents are needed to elucidate the theoretical basis and fully describe the algorithm. We detail all assumptions and approximations implemented to make the algorithm practical and their impacts on retrieval accuracy. These understandings point to future algorithm advances. Besides, significantly shortening the paper requires extensive effort to keep it coherent and readable. A minor revision is unlikely to get it done. We like to keep the contents and structures as-is.

2. Section 7.1: have you any suspects about distinct scatters/less correction between EPIC O3 and brewer at Paramaribo compared to other stations? I am wondering if it comes from either Brewer measurement uncertainties or algorithm retrieval artifacts in tropics. You shoud check this issue by performing the additional evaluation at stations adjacent to Paramaribo.

Response:

We show in Figure AC1 the EPIC-vs-Brewer $O_3$ scatter plot at the Mauna Loa station, which is in the tropic and provides measurements coincident with EPIC. Additionally, we include scatter plots between EPIC and MERRA-2 $O_3$ and between Brewer and MERRA-2 $O_3$ at the selected ground stations.

Figure AC1 shows lower $O_3$ correlations between EPIC and Brewer at Paramaribo and Mauna Loa. At these stations, $O_3$ correlations between EPIC and MERRA-2 and between Brewer and MERRA-2 are also low compared to those mid and high latitudes, indicating that the low correlation is likely common in the tropics and not specific to a particular station.

Correlations between MERRA-2 and EPIC are generally higher than between Brewer and EPIC and between Brewer and MERRA-2 at almost all the stations (except the South Pole station) shown in Figure AC1. MERRA-2 $O_3$ may be considered as improved OMI $O_3$. Therefore, MERRA-2 is more similar to EPIC as both are from the top of the atmosphere observations and have close spatial resolutions. Brewers are ground-based measurements, with different IFOVs and different measurement sensitivities that contribute to discrepancies between satellite and ground-based observations.

The lower correlations in the tropics than in other latitude bands are primarily due to its smaller dynamics $O_3$ range than those in higher latitude zones. The random EPIC $O_3$ error due to instrument noise is independent of the total column (see section 6.2.1). Adding this random component to $O_3$ with a narrower range degrades the correlation with the actual $O_3$ more than adding it to $O_3$ with a broader range. Furthermore, the random error in EPIC O3 decreases with higher viewing zenith angles when VZA < 70° (see section 6.2.1). Higher zenith angles are associated mainly with observations at higher latitudes. Thus, adding a smaller random component causes less degradation in correlation with actual $O_3$ at high latitudes. The lower correlations in the tropics do not represent higher uncertainties (i.e., lower precision and lower accuracy) of EPIC, as the biases and standard deviations in the low latitudes are smaller than those at high latitudes (see Figure AC1).

In short, both errors in EPIC $O_3$ and discrepancies in observations between EPIC and Brewer degrade their correlations in the tropics.

[Figure]

Figure AC1: Inter-comparison of total O₃ from EPIC, MERRA-2 (assimilated from OMI and MLS), and the ground-based Brewer spectrophotometers at 11 selected ground stations with high-cadence measurements.

---

## Author Comment (AC2)

We appreciate Referee #2's positive comments and careful review, which improves the paper. Below, Referee's comments are in blue and our responses in black.

**RC2**: 'Comment on amt-2022-156', Anonymous Referee #2, 30 Jun 2022
In this paper, the authors describe in a very comprehensive way the algorithm applied to DSCOVR observations to retrieve O3 and SO2 (for large volcanic eruptions) vertical columns using a direct-fitting approach.

The level of description of the algorithm is very high and unusual for publications, with the authors providing the basics of all involved processes in such a remote sensing application. Although this makes the paper quite long and perhaps not fully consistent with the editorial line, I find such papers useful for readers with less experience in the field.

The described algorithm itself is mature and provide high quality results and the derived retrievals are well characterized with solid error estimates. As the topic suits well AMT and I don't have major issues, I would recommend publication after the minor and technical corrections below have been considered.

**Minor comments:**

P. 9 lines 4-5: This statement is not clear to me. I understand that cloud/aerosol-free pixels have low LERs but why selecting such clean pixels only would remove the high VZA observations? Is the selection based on the LER values themselves or on independent cloud parameters?

Response:

From a polar-orbiting platform, a location on Earth may be observed with different VZAs (typically ranging from 0 to ~70 degrees) and a narrow range (~10 degrees) of SZAs during a calendar month. Since the reflectance of most natural surfaces increases with higher VZAs (e.g., see Coulson, 1966, Effects of Reflection Properties of Natural Surfaces in Aerial Reconnaissance, Appl. Opt. 5, 905-917, DOI: 10.1364/AO.5.000905) and furthermore IFOVs with higher VZAs (i.e., bigger footprint size) are more likely contaminated by clouds or aerosols, minimal LER selection tends to exclude observations with high VZAs.

Fig. 4b: How are those GLER values computed? Are they based on the Cox-Munk BRDF as well? The figure shows VZA dependences for low SZA but for EPIC, the SZA increases simultaneously with VZA. What's the influence of the SZA on GLER?

Response:

GLERs are inverted using Eq. (6) from simulated TOA radiances of a molecular atmosphere over a BRDF (Cox-Munk) surface.

[Figure]

Figure AC2: Similar to figure Fig. 4 of the manuscript, except the Sun at a higher zenith angle

The effect of a high SZA on GLER is illustrated in Figure AC2, which displays the GLERs for a high SZA ($\theta_s$=55˚). From the L1 point, EPIC does not observe from the directions close to specular reflections for high SZAs, while near the backward scattering directions, the GLERs are slightly elevated for high SZAs. In short, significantly elevated GLERs over water surfaces are not observed for high SZAs from EPIC. Figure R1(b) shows that the linear extrapolation of GLER at longer wavelengths yields highly accurate GLER estimations at shorter wavelengths for high SZAs.

Fig. 7 and P. 11 line 24: please specify which data base is used.

Response:

As described in the manuscript, the ice GLERs are constructed from Aura OMI and SNPP OMPS. We created this ice GLER climatology for use as a reference to calibrate reflective UV bands of polar-orbiting instruments (like NOAA-20 OMPS and S5P TROPOMI) and monitor their performances over time. The sample results in the manuscript are intended to illustrate that ice reflectivity is significantly anisotropic. We have not published this database, but we would share this ice GLER climatology with a reader who contacts the authors directly.

Fig. 10: the use of % is confusing here. Does it mean that what's plotted here is (I_RRS-I_ELA)/I_ELA X 100? If yes, please clarify. Otherwise, don't use %

Response:

We put in the figure caption $\varrho = \frac{I_{RRS} - I_{ELA}}{I_{ELA}} \times 100$.

P. 21 line 7-8: please explicit the granularity of the climatology.

Response:

Description changed to:  In short, M2TCO3 better captures the dynamical changes and spatiotemporal variations in $O_3$ profiles with higher resolutions in total $O_3$ column (25 DU), latitude (10°) and time (monthly).

Figure 15: Please comment on the large differences at high SZAs (edge of the disc)

Response:

Several versions of EPIC L1B data have been released since the launch of DSCOVR EPIC. The $O_3$ differences exhibited systematic changes in the interior of the disc between different L1B versions. However, near the disk edge, $O_3$ differences displayed large changes and even had sign reversion with calibration changes. Hence, the large differences near the edge of the disc are likely due to large discrepancies between measured and modeled radiances, given that higher calibration uncertainties of the edge pixels (see Cede et al., 2021, Raw EPIC Data Calibration, https://doi.org/10.3389/frsen.2021.702275) and large modeling errors at high zenith angles. Furthermore, retrievals from observations with large zenith angles (VZAs and/or SZAs) have considerably higher uncertainties due to enhanced sensitivities to other error sources (see the error analysis section), contributing to the large $O_3$ differences near the disc edge shown in Figure 15.

Algorithm 1/2 tables: Those tables are very useful. I think having flowcharts would be even nicer (keeping all references to Equations). Please consider doing this. Add also references to used data bases (minimum LER, cloud and snow parameters, O3/T° profiles).

Response:

We create flowcharts for $O_3$ and $SO_2$ retrievals, including references to L1, L2, ancillary, and climatological data used in making the L2 O3SO3AI product.

SO2 flagging: P. 33 line 33 and P. 341: it is not clear to me how "the vicinity outside the Delta_omega contour" and "adjacent areas" are defined. Please be more specific. Also I don't understand what is the reference value to draw the omega_1 contour, which is said to be taken between omega_min and omega_max. What does it mean? Do you take the mean of the two values or any other value?

Response:

We thank reviewer #2 for pointing out this unclear description.

After finding the $\Delta\Omega$ contour, we define an imaging region that covers the $\Delta\Omega$ contour. This rectangle region is formed by extending +/- 150 pixels from the contour, sufficiently large to cover volcanic plumes completely. Next, contour mapping of $\Omega_1$ is performed to find the area of $SO_2$ enhancements within the rectangle region, accomplished by stepping through the contour values from $\Omega_{max}$ to $\Omega_{min}$ to find the $\Omega$ value that yields the longest closed contour.

We have rewritten this part to describe the flagging procedure concisely and accurately.

P. 34 line 6: what is the justification to take as initial SO2 value the difference between two O3 columns (omega_1 and omega_2)

Response:

Based on the values of $O_3$ and $SO_2$ absorption cross-sections, one DU of $SO_2$ would cause 2 DU $\Omega_1$ and 0.7 DU $\Omega_2$ enhancements. Thus $\Delta\Omega=\Omega_1-\Omega_2$ is about 1.3 $\Xi$ (i.e., $\Xi = \Delta\Omega/1.3$). This estimate is accurate when the measurement sensitivities are the same for total $O_3$ and $SO_2$. But in general, they are different, with $SO_2$ sensitivity usually being lower than that of total $O_3$. In other words, one DU of $\Xi$ causes $\Delta\Omega$ that is less than 1.3 $\Xi$. For simplicity, $\Xi=\Delta\Omega$ is used as initial estimate, since the retrieved $\Xi$ is minimally affected by this initial estimate, as a loose constraint (i.e., $SO_2$ variance = $\Xi^2$) is imposed. The retrieved $\Xi$ is primarily determined by the radiance measurements.

P. 40 line 30: I don't think this is true that profile errors systematically increase for bright surfaces. In case of bright surfaces at ground level, the AK will be closer to 1 instead of having a strong decrease in sensitivity. So AKs will be much less altitude-dependent and errors due to the profile shape may be reduced.

Response:

We thank the reviewer for pointing the incorrect statement. We have added AKs in Figure 18 for a high reflectivity surface, showing AKs moving closer to 1 for high zenith angles and exceeding 1 in the troposphere for low zenith angles. These results imply that errors due to the profile shape decrease for high zenith angles and can change signs at low angles. In general, retrieval errors are reduced for high reflectivity surfaces. We have revised the manuscript to correct the incorrect description.

Error estimates: It would be beneficial to add up all error terms to have an estimate of the typical total errors. Of course, respective contributions vary significantly depending on the observation and geophysical conditions but I would suggest attempting to provide such total error estimates for (1) favourable (e.g. no cloud/aerosol, low angles (2) difficult conditions (high angles, aerosols).

Response:

We have added as a subsection to summarize the error estimates.

**Technical corrections :**

P. 1 line 18 : remove 'the' in 'located the between'

Done

P. 2 line 21 : add Metop-C

Metop-C added.

P. 2 line 29 : 'an LEO' --> 'a LEO'

Corrected.

P. 5 line 22 : define μ

In the manuscript, it was defined using $\theta_v = \cos^{-1}(\mu)$. Revised to $\mu = \cos(\theta_v)$

P. 14 line 2 : 'is a smooth' --> 'in a smooth' ?

The statement is rewritten as follows:

The change in $I_{TOA}$ due to the addition of aerosols and hence the cloud fraction ($f_c$) are smooth in wavelength.

P. 21 line 19 : should 'n' be 'p' instead for the number of e_k according Eq. 12 ?

'n' is replaced with 'p'.

P. 21 line 33 : O3 'climatology' instead of 'climatolgoy'

Corrected

P. 22 line 27 : suppress repetition of 'the'

Done

P. 24 line 16 : suppress repetition of 'the'

Done

P. 24 line 23: remove 'for as applicable'

Done

P. 24 line 25 : add Lerot et al., 2014

Reference added

P. 27 line 26 : close bracket after 'section 2.3'

Done

P. 36 line 12 : 'represent' instead of 'represents'

Corrected

Fig 23 : Expand the Y scale for the O3 differences to increase the readibility (+/- 15% instead of 30%)

Figure re-plotted with updated range ((+/- 15%).

P. 50 line 5 : rephrase the 'in this ATBD' in 'in this paper'

Done

P. 50 line 21 : 'laodings' --> 'loadings'

 Corrected